# Inhibition of SARS-CoV-2 polymerase by nucleotide analogs from a single-molecule perspective

Mona Seifert[1†], Subhas C Bera[1†], Pauline van Nies[1], Robert N Kirchdoerfer[2], Ashleigh Shannon[3], Thi-Tuyet-Nhung Le[3], Xiangzhi Meng[4], Hongjie Xia[5], James M Wood[6], Lawrence D Harris[6], Flavia S Papini[1], Jamie J Arnold[7], Steven Almo[8], Tyler L Grove[8], Pei-Yong Shi[9], Yan Xiang[4], Bruno Canard[3], Martin Depken[10], Craig E Cameron[7], David Dulin[1,11]*

[1]Junior Research Group 2, Interdisciplinary Center for Clinical Research, Friedrich-Alexander-University Erlangen-Nürnberg (FAU), Erlangen, Germany; [2]Department of Biochemistry and Institute of Molecular Virology, University of Wisconsin-Madison, Madison, United States; [3]Architecture et Fonction des Macromolécules Biologiques, CNRS and Aix-Marseille Université, Marseille, France; [4]Department of Microbiology, Immunology and Molecular Genetics, University of Texas Health Science Center at San Antonio, San Antonio, United States; [5]Department of Biochemistry and Molecular Biology, University of Texas Medical Branch, Galveston, United States; [6]The Ferrier Research Institute, Victoria University of Wellington, Wellington, New Zealand; [7]Department of Microbiology and Immunology, University of North Carolina School of Medicine, Chapel Hill, United States; [8]Department of Biochemistry, Albert Einstein College of Medicine, Bronx, Institute for Protein Innovation, Boston, United States; [9]Department of Biochemistry and Molecular Biology, University of Texas Medical Branch, Institute for Human Infections and Immunity, University of Texas Medical Branch, Sealy Institute for Vaccine Sciences, University of Texas Medical Branch, Sealy Center for Structural Biology & Molecular Biophysics, University of Texas Medical Branch, Galveston, United States; [10]Department of Bionanoscience, Kavli Institute of Nanoscience, Delft University of Technology, Delft, Netherlands; [11]Department of Physics and Astronomy, and LaserLaB Amsterdam, Vrije Universiteit Amsterdam, Amsterdam, Netherlands

*For correspondence:
d.dulin@vu.nl

[†]These authors contributed equally to this work

Competing interests: The authors declare that no competing interests exist.

**Abstract** The absence of 'shovel-ready' anti-coronavirus drugs during vaccine development has exceedingly worsened the SARS-CoV-2 pandemic. Furthermore, new vaccine-resistant variants and coronavirus outbreaks may occur in the near future, and we must be ready to face this possibility. However, efficient antiviral drugs are still lacking to this day, due to our poor understanding of the mode of incorporation and mechanism of action of nucleotides analogs that target the coronavirus polymerase to impair its essential activity. Here, we characterize the impact of remdesivir (RDV, the only FDA-approved anti-coronavirus drug) and other nucleotide analogs (NAs) on RNA synthesis by the coronavirus polymerase using a high-throughput, single-molecule, magnetic-tweezers platform. We reveal that the location of the modification in the ribose or in the base dictates the catalytic pathway(s) used for its incorporation. We show that RDV incorporation does not terminate viral RNA synthesis, but leads the polymerase into backtrack as far as 30 nt, which may appear as termination in traditional ensemble assays. SARS-CoV-2 is able to evade the endogenously synthesized product of the viperin antiviral protein, ddhCTP, though the polymerase incorporates

this NA well. This experimental paradigm is essential to the discovery and development of therapeutics targeting viral polymerases.

## Introduction

SARS-CoV-2 has infected hundreds of million humans worldwide, causing millions of deaths, with numbers still on the rise. We are currently living through the third coronavirus outbreak in less than 20 years, and we are desperately in need of broad-spectrum antiviral drugs that are capable of targeting this emerging family of human pathogens. To this end, nucleotide analogs (NAs) represent a powerful approach, as they target the functionally and structurally conserved coronavirus polymerase, and their insertion in the viral RNA induces either premature termination or a lethal increase in mutations. The coronavirus polymerase is composed of the nsp12 RNA-dependent RNA polymerase (RdRp), and the nsp7 and nsp8 co-factors, with a stoichiometry of 1:1:2 (*Kirchdoerfer and Ward, 2019*; *Hillen et al., 2020*; *Gao et al., 2020*; *Wang et al., 2020*). This polymerase is thought to associate with several additional viral proteins, including the nsp13, a 5′-to-3′ RNA helicase (*Chen et al., 2020*; *Yan et al., 2020*), and the nsp14, a 3′-to-5′ exoribonuclease (*Agostini et al., 2018*; *Bouvet et al., 2012*; *Ferron et al., 2018*; *Ogando et al., 2019*; *Subissi et al., 2014*; *Eckerle et al., 2007*). The latter proofreads the terminus of the nascent RNA following synthesis by the polymerase and associated factors (*Robson et al., 2020*), a unique feature of coronaviruses relative to all other families of RNA viruses. Proofreading likely contributes to the stability of the unusually large, ~30 kb, coronavirus genome. In addition, proofreading may elevate the tolerance of coronaviruses to certain NAs (e.g., ribavirin; *Smith et al., 2013*) and therefore should be considered in the development of potent NAs. In other words, nsp14 adds another selection pressure on NAs: not only they must be efficiently incorporated by nsp12, they must also evade detection and excision by nsp14. Remdesivir (RDV) is a recently discovered NA that showed efficacy against Ebola infection (*Siegel et al., 2017*) and has been successfully repurposed for the treatment of SARS-CoV-2 infection (*Agostini et al., 2018*; *Gordon et al., 2020a*; *Gordon et al., 2020b*; *Pruijssers and Denison, 2019*; *Jockusch et al., 2020*; *Chien et al., 2020*). The success of RDV relies on its efficient incorporation by the polymerase (*Gordon et al., 2020b*; *Dangerfield et al., 2020*) and probable evasion of the proofreading machinery (*Agostini et al., 2018*). Understanding how RDV achieves these two tasks, will help to guide the rational design of more efficacious NAs for the current and future outbreaks. To this end, it is essential to build a comprehensive model describing the selection and incorporation mechanisms that control the utilization of NAs by the coronavirus polymerase and to define the determinants of the base and ribose responsible for selectivity and potency. We have therefore compared several analogs of the same natural nucleotide to determine how the nature of the modifications changes selection/mechanism of action. Magnetic tweezers permit the dynamics of an elongating polymerase/polymerase complex to be monitored in real time and the impact of NAs to be monitored in the presence of all four natural nucleotides in their physiological concentration ranges. Here, we present a magnetic tweezers assay to provide insights into the mechanism and efficacy of current and under-explored NAs on the coronavirus polymerase.

## Results

### Monitoring SARS-CoV-2 RNA synthesis at the single-molecule level

To enable the observation of rare events, such as nucleotide mismatch and NA incorporation, even in the presence of saturating NTP concentration, we have developed a single-molecule, high-throughput, magnetic tweezers assay to monitor SARS-CoV-2 RNA synthesis activity at near single base resolution (*Dulin et al., 2017*). A SARS-CoV-2 polymerase formed of nsp12, nsp7, and nsp8 (*Figure 1—figure supplement 1A*) assembles and initiates RNA synthesis at the 3′-end of the magnetic bead-attached handle, and converts the 1043 nt long single-stranded (ss) RNA template into a double-stranded (ds) RNA in the presence of NTPs and at constant force, that is, 35 pN if not mentioned otherwise (*Figure 1—figure supplement 1B–D*; see Materials and methods). The conversion from ssRNA to dsRNA displaces the magnetic bead along the vertical axis and directly informs on the number of incorporated nucleotides (*Figure 1A*, see Materials and methods; *Dulin et al., 2015a*). During each experiment, hundreds of magnetic beads are followed in parallel (*Figure 1—*

**eLife digest** To multiply and spread from cell to cell, the virus responsible for COVID-19 (also known as SARS-CoV-2) must first replicate its genetic information. This process involves a 'polymerase' protein complex making a faithful copy by assembling a precise sequence of building blocks, or nucleotides.

The only drug approved against SARS-CoV-2 by the US Food and Drug Administration (FDA), remdesivir, consists of a nucleotide analog, a molecule whose structure is similar to the actual building blocks needed for replication. If the polymerase recognizes and integrates these analogs into the growing genetic sequence, the replication mechanism is disrupted, and the virus cannot multiply. Most approaches to study this process seem to indicate that remdesivir works by stopping the polymerase and terminating replication altogether. Yet, exactly how remdesivir and other analogs impair the synthesis of new copies of the virus remains uncertain.

To explore this question, Seifert, Bera et al. employed an approach called magnetic tweezers which uses a magnetic field to manipulate micro-particles with great precision. Unlike other methods, this technique allows analogs to be integrated under conditions similar to those found in cells, and to be examined at the level of a single molecule.

The results show that contrary to previous assumptions, remdesivir does not terminate replication; instead, it causes the polymerase to pause and backtrack (which may appear as termination in other techniques). The same approach was then applied to other nucleotide analogs, some of which were also found to target the SARS-CoV-2 polymerase. However, these analogs are incorporated differently to remdesivir and with less efficiency. They also obstruct the polymerase in distinct ways.

Taken together, the results by Seifert, Bera et al. suggest that magnetic tweezers can be a powerful approach to reveal how analogs interfere with replication. This information could be used to improve currently available analogs as well as develop new antiviral drugs that are more effective against SARS-CoV-2. This knowledge will be key at a time when treatments against COVID-19 are still lacking, and may be needed to protect against new variants and future outbreaks.

figure supplement 1E), yielding dozens of traces of SARS-CoV-2 polymerase activity per experiment (*Figure 1B*, *Figure 1—figure supplement 2A*). As previously observed for other viral RdRps (*Dulin et al., 2017*; *Dulin et al., 2015a*; *Seifert et al., 2020*), the traces reveal substantial, heterogeneous dynamics, with bursts of activity interrupted by pauses of duration varying from ~0.5 s to ~60 s in *Figure 1B*. This dynamic is intrinsic to the polymerase elongation kinetics and does not result from viral proteins exchange (*Bera et al., 2021*). To extract the elongation kinetics of the SARS-CoV-2 polymerase, we scanned the traces with non-overlapping 10-nt windows to measure the duration of time required to complete the 10 successive nucleotide-incorporation events. Each duration of time has been coined a dwell time, which is the kinetic signature of the rate-limiting event of the 10 nt addition, that is, the 10 nucleotide addition cycles themselves, or a pause (*Dulin et al., 2017*; *Dulin et al., 2015a*; *Seifert et al., 2020*; see Materials and methods). We fitted the distribution of dwell times using the stochastic-pausing model that describes well the kinetics of nucleotide addition of the coronavirus polymerase (*Bera et al., 2021*) and other viral RdRps (*Dulin et al., 2017*; *Dulin et al., 2015a*; *Seifert et al., 2020*; *Dulin et al., 2015b*; *Figure 1C*; see Materials and methods). This model is composed of four distributions: a pause-free nucleotide addition rate, Pause 1, Pause 2, and the backtrack pauses (*Figure 1C*; see Materials and methods), and fit parameters values extracted from triplicate experiments fall within statistical errors (*Figure 1—figure supplement 2B,C*). Statistics and all parameter values extracted from the analysis are reported in *Supplementary file 1* and *Supplementary file 2*. SARS-CoV-2 polymerase elongation kinetics is well described by a robust model where the nucleotide addition rate is the kinetic signature of the nucleotide addition burst (NAB) pathway, from which the RdRp stochastically and rarely switches into the slow nucleotide addition (SNA) pathway, and even more rarely into the very slow nucleotide addition (VSNA) pathway, the latter being consistent in rate and probability with mismatch incorporation (*Figure 1D*; *Bera et al., 2021*) Pause 1 and Pause 2 are respectively the kinetic signatures of the SNA and VSNA pathways (*Figure 1D*), while the long-lived pauses relate to a catalytically

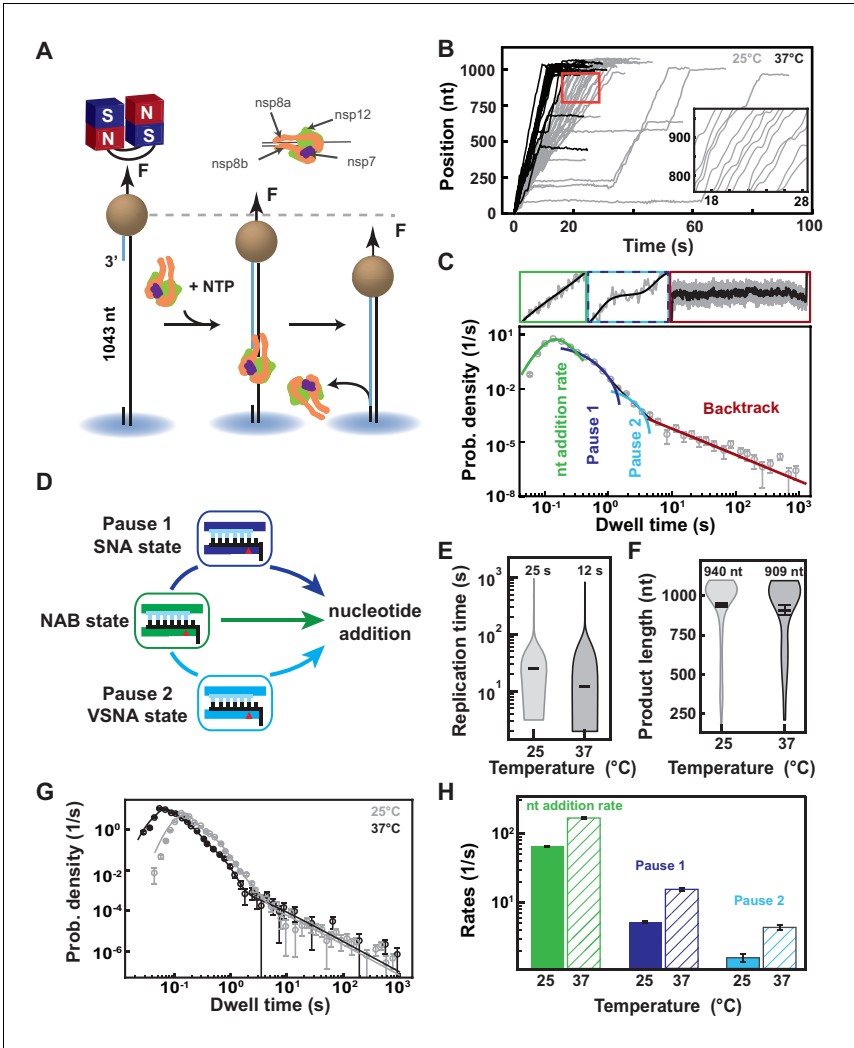

**Figure 1.** SARS-CoV-2 polymerase is a fast and processive RNA polymerase complex. (**A**) Schematic of the magnetic tweezers assay to monitor RNA synthesis by the SARS-CoV-2 polymerase complex. A magnetic bead is attached to a glass coverslip surface by a 1043 long ssRNA construct that experiences a constant force F (35 pN if not mentioned otherwise). The polymerase, formed by nsp7, nsp8, and nsp12, assembles at the 3′-end of the RNA strand annealed to the template. The subsequent conversion of the ssRNA template into dsRNA reduces the end-to-end extension of the tether, signaling replication activity. (**B**) SARS-CoV-2 polymerase activity traces acquired at either 25°C (gray) or 37°C (black), showing bursts of nucleotide addition interrupted by pauses. The inset is a zoom-in of the traces captured in the red square. (**C**) The dwell times collected from (**B**) are assembled into a distribution that is fitted using a stochastic pausing model (see Materials and methods; solid lines). The model includes four different probability distribution functions that describe the event that kinetically dominates the dwell time: uninterrupted 10 nucleotide additions (green), exponentially distributed Pause 1 and Pause 2 (blue and cyan, respectively), and the power-law distributed backtrack (red). (**D**) The dwell time distribution in (**C**) is described by the viral RdRp kinetic model (adapted from *Dulin et al., 2017*). Fast nucleotide addition is achieved by the nucleotide addition burst (NAB) pathway with the nucleotide addition rate extracted from (**C**). Pause 1 and Pause 2 are the kinetic signatures of the slow and very slow nucleotide addition (SNA and VSNA, respectively) pathways, the latter being likely related to nucleotide mismatch incorporation. (**E**) Total replication time and (**F**) product length of SARS-CoV-2 polymerase activity traces at either 25°C or 37°C. The median total replication time and the mean product length are indicated above the violin plots, and represented as thick horizontal lines. The error bars represent one standard deviation extracted from 1000 bootstraps. (**G**) Dwell time distributions of SARS-CoV-2 polymerase activity traces at 25°C (gray circles) and 37°C (black circles) extracted from (**B**), and their respective fit to the stochastic-pausing model (corresponding solid lines). (**H**) Nucleotide addition rate (green), Pause 1 (dark blue), and Pause 2 (cyan) exit rates at either 25°C or 37°C (solid and hatched bars, respectively) extracted from (**G**). The

*Figure 1 continued on next page*

*Figure 1 continued*

error bars in (**C** and **G**) represent one standard deviation extracted from 1000 bootstraps. The error bars in (**H**) are one standard deviation extracted from 100 bootstraps.

The online version of this article includes the following source data and figure supplement(s) for figure 1:

**Figure supplement 1.** Experimental conditions of SARS-CoV-2 polymerase high throughput magnetic tweezers experiments.

**Figure supplement 1—source data 1.** Source image for the SDS-PAGE gel in *Figure 1—figure supplement 2A*.

**Figure supplement 2.** Selection of SARS-CoV-2 polymerase elongation traces and reproducibility.

incompetent polymerase backtrack state, that is, the polymerase diffuses backward on the template strand leading the product strand 3′-end to unwind and exit via the NTP channel without cleavage (see Materials and methods *Bera et al., 2021*; *Malone et al., 2021*). Increasing the temperature from 25°C to 37°C, SARS-CoV-2 polymerase reveals a strong temperature dependence, which translates into a twofold decrease in the median replication time (*Figure 1E*), while not affecting the RNA synthesis product length (*Figure 1F*). Analyzing the dwell time distribution at 25°C and 37°C (*Figure 1G*), we extracted a ~2.6-fold enhancement in nucleotide addition rate, from $(65.6 \pm 0.5)\,\mathrm{nt.s^{-1}}$ to $(169.0 \pm 3.8)\,\mathrm{nt.s^{-1}}$, making the SARS-CoV-2 polymerase the fastest RNA polymerase characterized to date (*Figure 1H*; *Dangerfield et al., 2020*; *Shannon et al., 2020a*). Pause 1 and Pause 2 exit rates also increased by ~3-fold (*Figure 1H*), whereas their respective probabilities increased by twofold and fivefold (*Figure 1—figure supplement 2D*). The latter results are rather surprising, as poliovirus and human rhinovirus C RdRps showed only an exit rate increase with no change in probability (*Seifert et al., 2020*).

## 3′-dATP versus remdesivir-TP: both ATP competitors but two different modes of incorporation

Next, we investigated how the elongation kinetics and the product length of SARS-CoV-2 polymerase were affected by two adenosine analogs, 3′-dATP and RDV-TP (*Figure 2—figure supplement 1*). 3′-dATP is an obligatory terminator of RNA chain elongation for viral RdRp (*Gohara et al., 2004*). RDV-TP has also been suggested to cause chain termination but only several cycles of nucleotide addition after its incorporation. If this is the true mechanism of action, then the experimental outcome of the presence of any of these two analogs should be indistinguishable in our assay.

In the presence of 500 µM NTP and 500 µM 3′-dATP, the ability of the SARS-CoV-2 polymerase to reach the end of the template (1043 nt) was compromised (*Figure 2A* vs. *Figure 1B*). Indeed, increasing 3′-dATP concentration up to 2000 µM, only reduced the mean product length of SARS-CoV-2 polymerase by ~1.7-fold, from $(940 \pm 13)\,\mathrm{nt}$ to $(566 \pm 33)\,\mathrm{nt}$ (mean±standard deviation) (*Figure 2B*), while not affecting the kinetics of RNA synthesis (*Figure 2C*, *Figure 2—figure supplement 2A–C*).

We derived a model to determine the effective incorporation rate $\gamma$, that is, the average number of nucleotide addition cycles before terminator incorporation at equimolar concentration of competing natural nucleotide (in the presence of all NTPs) (see Materials and methods, *Figure 2—figure supplement 3*). This model fits very well to the mean product length as a function of 3′-dATP:ATP stoichiometry (*Figure 2B*), for example, $\gamma_{3'-dATP,\,500\,\mu\mathrm{M\,ATP}} = (780 \pm 64)\,\mathrm{nt}$, meaning that the polymerase incorporates on average 780 nt before incorporating one 3′-dATP and terminating RNA synthesis (see Materials and methods). A subsaturating concentration of NTP increases the probability to enter both the SNA and the VSNA pathways (*Figure 1D*; *Bera et al., 2021*), that is, Pause 1 and Pause 2 probability, and would increase the effective incorporation rate of 3′-dATP, providing it is incorporated via any of the two SNA states. By decreasing ATP concentration from 500 µM to 50 µM, we indeed observed an increase in Pause 1 and Pause 2 probabilities by more than twofold and threefold, from $(0.060 \pm 0.002)$ to $(0.149 \pm 0.005)$ and from $(0.0033 \pm 0.0009)$ to $(0.0115 \pm 0.0026)$, respectively (*Figure 2—figure supplement 2A–F*). Adding 500 µM of 3′-dATP significantly shortened the traces in comparison to the 500 µM ATP condition (*Figure 2A,D*). However, the effective incorporation rate of 3′-dATP was identical at both concentrations of ATP, that is, $\gamma_{3'-dATP,\,50\,\mu M\,ATP} = (777 \pm 50)\,\mathrm{nt}$ (*Figure 2E*), which indicates that 3′-dATP incorporation is only driven by stoichiometry, despite the significant increase in the SNA (Pause 1) and VSNA (Pause 2) pathways

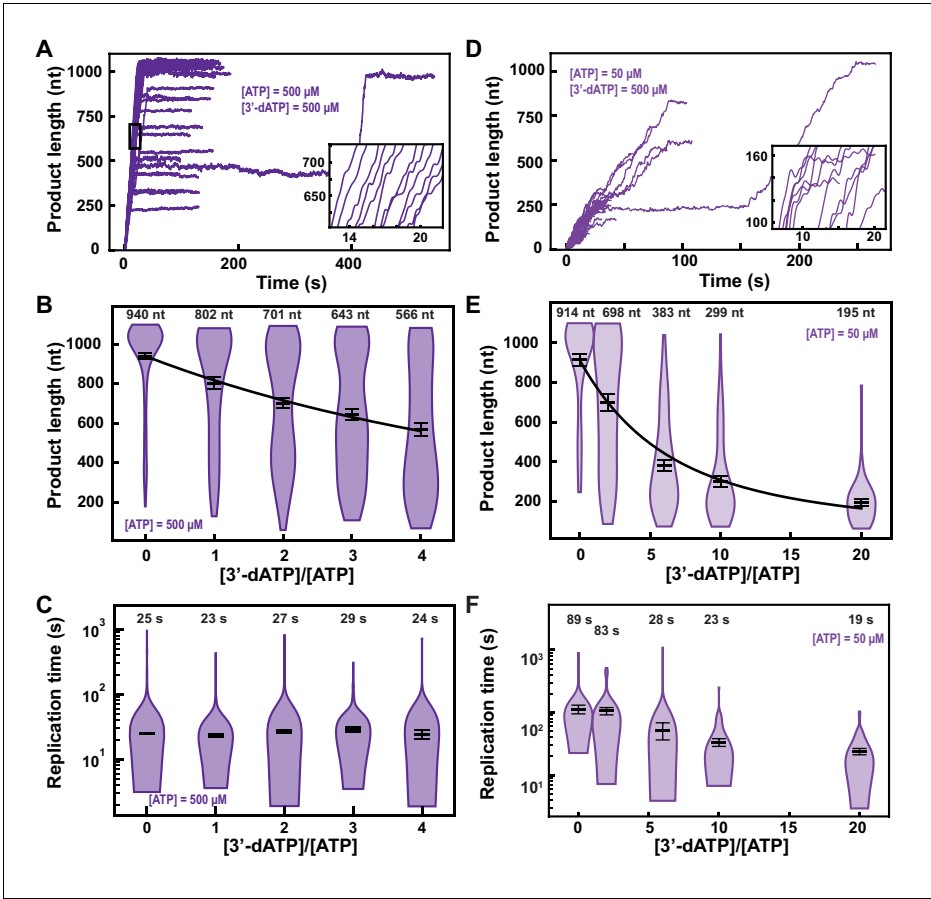

**Figure 2.** 3′-dATP is an effective chain terminator for the SARS-CoV-2 polymerase. (**A**) SARS-CoV-2 replication traces for 500 µM NTPs and 500 µM 3′-dATP; and (**D**), for 50 µM ATP, 500 µM all other NTPs and 500 µM 3′-dATP. (**B, E**) SARS-CoV-2 polymerase product length for the 1043 nt long template using the indicated concentration of ATP, 500 µM of other NTPs, as a function of [3′-dATP]/[ATP]. The mean values are indicated above the violin plots, and represented by horizontal black thick lines flanked by one standard deviation error bars extracted from 1000 bootstraps. (**C, F**) Replication time for the reaction conditions described in (**B, E**). The medians are indicated above the violin plots, and represented by horizontal black thick lines flanked by one standard deviation error bars extracted from 1000 bootstraps. In (**B, E**), the solid lines are the fits of the terminator effective incorporation rate (see Materials and methods). In (**A, D**), the insets are zoom-in of the replication traces captured in the black square.

The online version of this article includes the following figure supplement(s) for figure 2:

**Figure supplement 1.** Structure of the nucleotide analogs used in this study.

**Figure supplement 2.** SARS-CoV-2 polymerase activity traces kinetics in presence of 3′-dATP.

**Figure supplement 3.** Probabilistic model describing the competition for incorporation of a nucleic acid chain terminator NA and natural nucleotide.

**Figure supplement 4.** Decreasing the applied tension does not change the effect of nucleotide analogs (NAs) on the SARS-CoV-2 polymerase elongation.

probabilities. Therefore, we conclude that 3′-dATP utilizes the NAB pathway for incorporation (*Figure 1D*). Of note, the decrease in the median replication time is due to the shortening of the product length from early termination (*Figure 2F*). Replicating the experiment at a 3′-dATP:ATP stoichiometry of 6 but now at 25 pN showed no significant differences in final product length in comparison to the data acquired at 35 pN (*Figure 2E*, *Figure 2—figure supplement 4A*).

RDV-TP is an adenine analog with a 1′-cyano modification that has recently been shown to outcompete ATP for incorporation (*Gordon et al., 2020b*; *Dangerfield et al., 2020*; *Figure 2—figure supplement 1*), while exhibiting a low cytotoxicity (*Pruijssers et al., 2020*). RDV-TP has been proposed to induce delayed chain termination at i+3 (i being RDV incorporation position) during RNA

synthesis by the core polymerase (*Gordon et al., 2020a*; *Gordon et al., 2020b*). Adding 100 μM RDV-TP in a reaction buffer containing 500 μM NTPs showed a dramatic increase in the pause density and duration, but most of the traces reached the end of the template (*Figure 3A*). We indeed observed a final product length largely unaffected at all concentrations of RDV-TP (*Figure 3B*), while the median time for RNA synthesis increased by more than tenfold (*Figure 3C*), for RDV-TP concentrations increasing up to 300 μM. Therefore, the RDV-TP mechanism of action is not termination. We then investigated the origin of the pause induced by RDV-TP incorporation using our stochastic-pausing model (*Figure 3D*, *Figure 3—figure supplement 1B*). While the nucleotide addition rate is unaffected by RDV-TP, all pauses are significantly impacted. The exit rates of Pause 1 and Pause 2 decreased by fourfold and tenfold (*Figure 3E*), while their probabilities increased by twofold and fourfold, respectively (*Figure 3F*). Most notably, the backtrack pause probability increased by 28-fold, from $(0.0005 \pm 0.0001)$ to $(0.0142 \pm 0.0015)$, when increasing RDV-TP concentration up to 300 μM. The backtrack pause probability increase was such that it most likely affected the probability and the exit rates of Pause 1 and Pause 2 above 50 μM RDV-TP (*Figure 3F*).

As expected, the almost identical SARS-CoV-1 polymerase (*Kirchdoerfer and Ward, 2019*) demonstrated a similar kinetic signature to RDV-TP incorporation (*Figure 3—figure supplement 2A–F*, *Supplementary file 1*), though to a lesser extent, for example, the backtrack probability increased by ~9-fold when raising RDV-TP concentration up to 300 μM versus 28-fold for SARS-CoV-2 polymerase.

To verify whether the applied tension modifies the incorporation kinetics of RDV-TP by the SARS-CoV-2 polymerase, we replicated the experiment using 500 μM NTP and 100 μM RDV-TP at 25 pN, that is, a 10 pN lower force (*Figure 2—figure supplement 4B*). We did not observe any significant difference between the two experiments at 35 pN and 25 pN (*Figure 2—figure supplement 4C,D*), indicating that tension does not play a significant role in RDV-TP incorporation.

Using our recently developed ultra-stable magnetic tweezers (*Bera et al., 2021*), we wanted to directly monitor polymerase backtrack induced by RDV-TP incorporation. To this end, we performed an experiment with 10 μM RDV-TP and 50 μM ATP, keeping all other NTPs at 500 μM (*Figure 3GH*, *Figure 3—figure supplement 1A*). We hypothesized that lowering the concentration of ATP, the natural competitor of RDV, would increase the incorporation yield of RDV and therefore polymerase backtrack probability. A close observation of the longest-lived pauses clearly demonstrates polymerase backtrack, as deep as ~30 nt (*Figure 3GH*, *Figure 3—figure supplement 1A*), demonstrating that RDV incorporation induces polymerase backtrack, which leads to long-lived pauses.

To verify whether the incorporation of RDV-TP is stoichiometric, we further analyzed the experiment performed at 50 μM ATP, 500 μM all other NTPs, and 10 μM RDV-TP, at 25°C and 35 pN (*Figure 3—figure supplement 3A*) (coined low ATP and RDV-TP concentrations), that is, the same stoichiometry as 500 μM all NTPs and 100 μM RDV-TP (coined high ATP and RDV-TP concentrations). In absence of RDV-TP, the decrease in ATP concentration from 500 μM to 50 μM increased dramatically Pause 1 and Pause 2 probability by 2.5-fold and 3.5-fold, respectively, while the backtrack pause remained unchanged. The large increase in both Pause 1 and Pause 2 probabilities further disentangled the distribution of these pauses from the backtrack pause, and we therefore did not expect a strong crossover of the latter on the former (as observed at 500 μM all NTPs). We noticed an average product length of $(633 \pm 30)\,\text{nt}$ at low ATP and RDV-TP concentrations, that is, a ~30% shorter than for any other conditions presented in *Figure 3—figure supplement 3B*, even though we acquired data for a much longer duration than at high ATP and RDV-TP concentrations, that is, 11,000 s vs. 1,600 s, respectively. Interestingly, this result resembles what was observed at a 3′-dATP:ATP stoichiometry of ~3 (*Figure 2B*), indicating that RDV-TP induces what resembles termination at low ATP concentration. We also observed a ~2.3-fold longer median replication time than at high ATP and RDV-TP concentrations (*Figure 3—figure supplement 3C*), an increase largely underestimated as a large fraction of the traces never reached the end of the template during the measurement (*Figure 3—figure supplement 3B*). Applying the stochastic-pausing model to the dwell time distribution of the low ATP and RDV-TP concentrations data (*Figure 3—figure supplement 3D*), we found the nucleotide addition rate unchanged, while Pause 1 and Pause 2 exit rates were lower than in absence of RDV-TP, that is, by 1.4-fold and 2.3-fold, respectively (*Figure 3—figure supplement 3E*). At low ATP concentration, the probabilities of Pause 1 and Pause 2 were largely unaffected by the presence of RDV-TP, similarly to what was observed at 37°C (*Figure 3—figure supplement 3F*). Most remarkably, the backtrack pause probability increased dramatically at

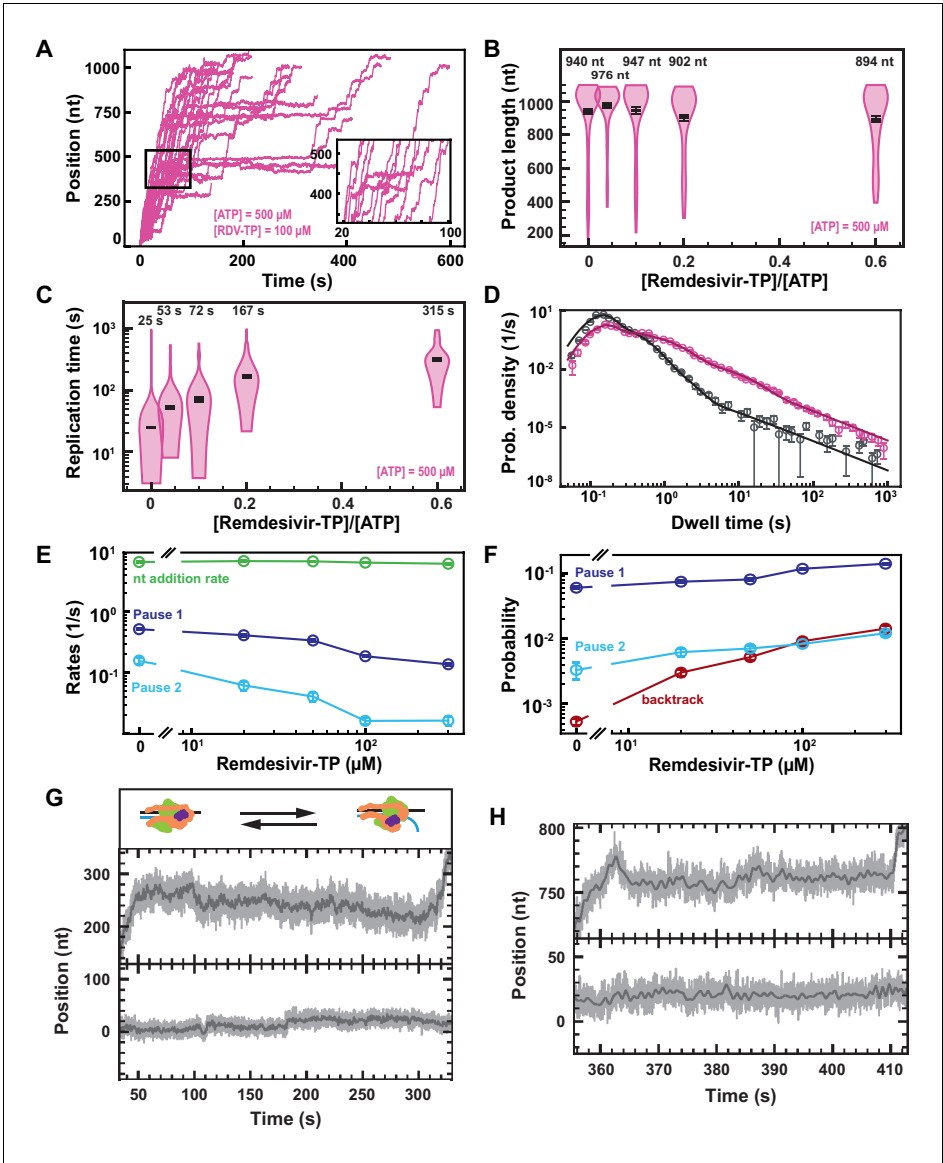

**Figure 3.** Remdesivir-TP (RDV-TP) is not a chain terminator but induces long-lived SARS-CoV-2 polymerase backtrack. (**A**) SARS-CoV-2 polymerase activity traces for 500 μM NTPs and 100 μM RDV-TP. The inset is a zoom-in of the polymerase activity traces captured in the black square. (**B**) SARS-CoV-2 polymerase product length for the 1043 nt long template using the indicated concentration of ATP, 500 μM of other NTPs, as a function of [RDV-TP]/[ATP]. The mean values are indicated above the violin plots, and represented by horizontal black thick lines flanked by one standard deviation error bars extracted from 1000 bootstraps. (**C**) Replication time for the reaction conditions described in (**B**). The median values are indicated above the violin plots, and represented by horizontal black thick lines flanked by one standard deviation error bars extracted from 1000 bootstraps. (**D**) Dwell time distributions of SARS-CoV-2 polymerase activity traces for 500 μM NTPs in the absence (gray) or presence of 100 μM RDV-TP (pink). The corresponding solid lines are the fit of the stochastic-pausing model. (**E**) Nucleotide addition rate (green), Pause 1 (dark blue), and Pause 2 (cyan) exit rates for [NTPs]=500 μM and several RDV-TP concentrations. (**F**) Probabilities to enter Pause 1 (dark blue), Pause 2 (cyan), and the backtrack (red) states for the conditions described in (**E**). The error bars in (**D**) represent one standard deviation extracted from 1000 bootstraps and the error bars in (**E, F**) represent one standard deviation extracted from 100 bootstrap procedures. (**G, H**) Examples of deep SARS-CoV-2 backtracks induced by RDV-TP incorporation (top) and traces showing no polymerase activity (bottom). Traces acquired using ultra-stable magnetic tweezers as described in *Bera et al., 2021* at 35 pN, 58 Hz acquisition frequency (gray), low-pass filtered at 1 Hz (dark gray), and using a SARS-CoV-2 polymerase reaction buffer containing 10 μM RDV-TP, 50 μM ATP, and 500 μM all other NTPs. The insert in (**G**) shows a schematic of backtracking polymerase.

*Figure 3 continued on next page*

*Figure 3 continued*

The online version of this article includes the following figure supplement(s) for figure 3:

**Figure supplement 1.** SARS-CoV-2 polymerase elongation traces in presence of RDV-TP at 25°C.

**Figure supplement 2.** SARS-CoV-1 polymerase activity traces kinetics in presence of RDV-TP.

**Figure supplement 3.** Lower ATP concentration at constant RDV-TP:ATP stoichiometry increases the effects of RDV-TP on SARS-CoV-2 polymerase elongation kinetics.

**Figure supplement 4.** SARS-CoV-2 polymerase activity traces kinetics in presence of RDV-TP at 37°C.

---

low ATP and RDV-TP concentrations, even more so than at high ATP and RDV-TP concentrations, that is, 43-fold versus 18-fold, respectively (*Figure 3—figure supplement 3F*). The main effect of RDV-TP is to increase the backtrack pause probability. In our previous study of the impact of T-1106-TP on poliovirus RdRp, we showed that T-1106 incorporation induces long-lived backtrack pauses that appear as termination in ensemble assays (*Dulin et al., 2017*). Interestingly, lowering ATP concentration increases the potency of RDV-TP by dramatically increasing the backtrack pause probability. However, we know such a pause is catalytically incompetent (*Bera et al., 2021*), and therefore the increase of the backtrack probability is an illustration of the effect of RDV-TP incorporation at low nucleotide concentration: the increased energy barrier induced by the steric clash of RDV-TP with the nsp12 serine-861 reduces dramatically the likelihood of a successful forward translocation of the polymerase (*Gordon et al., 2020b*; *Kokic et al., 2021*). This likelihood is even further reduced at low NTP concentration, which dramatically increases the probability of polymerase backtrack (*Dulin et al., 2015c*).

We previously observed that increasing the temperature helped to further disentangle the distributions of the different pauses (*Seifert et al., 2020*). We therefore performed an experiment at 37°C in the presence of 100 µM RDV-TP and 500 µM all NTPs (*Figure 3—figure supplement 4A*). The nucleotide addition rate significantly increased with temperature, while this increase was not affected by the presence of RDV-TP (*Figure 1H*, *Figure 3—figure supplement 4B*). On the one hand, Pause 1 and Pause 2 exit rates significantly decreased by threefold and ninefold, respectively, when the reaction was performed with RDV-TP (*Figure 3—figure supplement 4B*). On the other hand, Pause 1 and Pause 2 probabilities were unaffected by the presence of RDV-TP (*Figure 3—figure supplement 4C*), supporting the notion that the increase in probability in the experiments performed at 25°C was the consequence of the polymerase backtrack pause distribution biasing Pause 1 and Pause 2 distributions (*Dulin et al., 2017*). The backtrack pause probability still increased by more than sevenfold, that is, from $(0.0003 \pm 0.0001)$ to $(0.0022 \pm 0.0007)$. The lesser increase in the backtrack pause probability at 37°C (28-fold at 25°C) is consistent with a model where RDV-MP represents a barrier to translocation, which crossing would be facilitated by increasing the thermal energy.

If RDV-TP incorporation resulted in a pause of similar exit rates as Pause 1 and Pause 2, but not mechanistically related to them, we would expect an increase in the probabilities of both pauses. However, in conditions where Pause 1 and Pause 2 distribution were clearly distinguishable from the backtrack pause distribution when having RDV-TP in the reaction buffer, that is, at 37°C and at low ATP concentration, we did not observe an increase in Pause 1 and Pause 2 probabilities. Therefore, we suggest that RDV-TP is incorporated by the SNA and VSNA pathways (*Bera et al., 2021*), leading to polymerase backtrack when failing at overcoming the increased energy barrier resulting from the clash of RDV-MP with serine-861.

## T-1106-TP is incorporated with a low probability via the VSNA state

Pyrazine-carboxamides represent a promising family of antiviral NAs, of which the best-known member is Favipiravir (T-705), recently approved to treat influenza virus infection (*Furuta et al., 2009*), and considered against SARS-CoV-2. We studied here another member of this family, T-1106 triphosphate (T-1106-TP), which is chemically more stable than T-705, while presenting similar antiviral properties (*Dulin et al., 2017*; *Shannon et al., 2020b*). T-1106-TP competes for incorporation against ATP and GTP in a sequence-dependent manner (*Dulin et al., 2017*; *Shannon et al., 2020b*). Adding 500 µM of T-1106-TP in a reaction buffer containing 500 µM NTPs significantly increased the number and duration of pauses observed in SARS-CoV-2 RNA synthesis activity traces (*Figure 4A*),

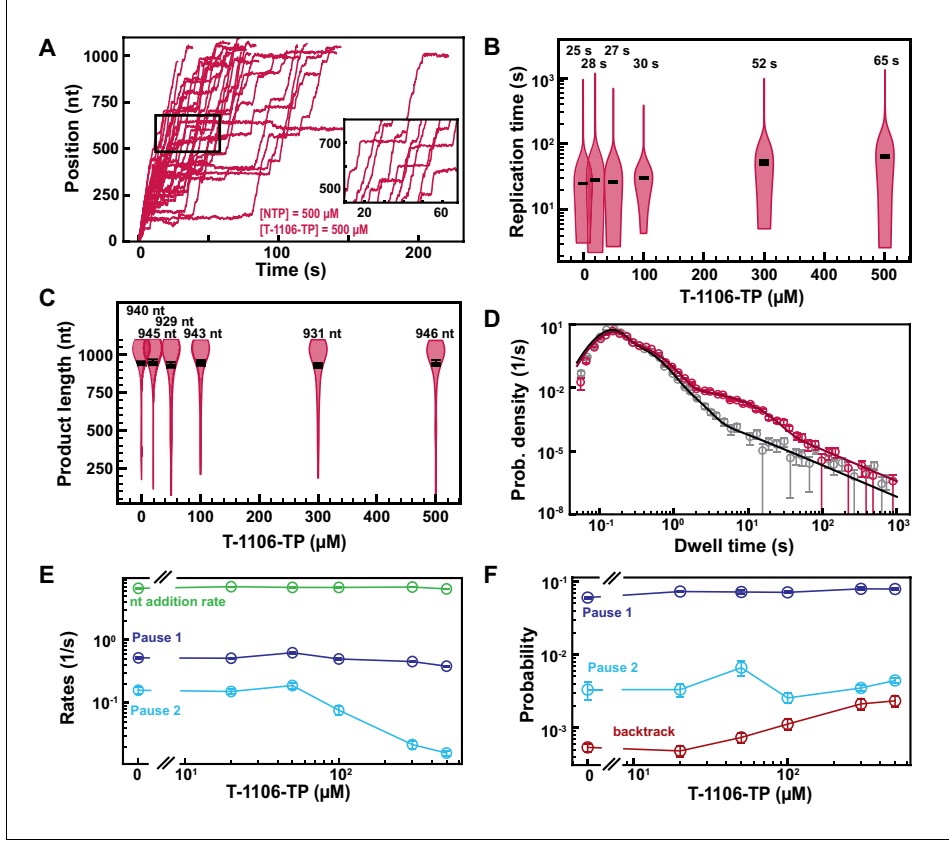

**Figure 4.** T-1106-TP incorporation induces pauses of intermediate duration and backtrack. (**A**) SARS-CoV-2 polymerase activity traces in the presence of 500 µM NTPs, in the presence of 500 µM T-1106-TP. The inset is a zoom-in of the polymerase activity traces captured in the black square. (**B**) SARS-CoV-2 replication time for the 1043 nt long template using 500 µM of all NTPs, and the indicated concentration of T-1106-TP. The median values are indicated above the violin plots, and represented by horizontal black thick lines flanked by one standard deviation error bars extracted from 1000 bootstraps. (**C**) SARS-CoV-2 polymerase product length using 500 µM NTPs and the indicated concentration of T-1106-TP. The mean values are indicated above the violin plots, and represented by horizontal black thick lines flanked by one standard deviation error bars extracted from 1000 bootstraps. (**D**) Dwell time distributions of SARS-CoV-2 polymerase activity traces for 500 µM NTP either without (gray) or with 500 µM (red) T-1106-TP. The corresponding solid lines are the fit to the stochastic-pausing model. (**E**) Nucleotide addition rate (green), Pause 1 (dark blue), and Pause 2 (cyan) exit rates for [NTPs]=500 µM and several T-1106-TP concentrations. (**F**) Probabilities to enter Pause 1 (dark blue), Pause 2 (cyan), and the backtrack (red) states for the conditions described in (**E**). The error bars denote one standard deviation from 1000 bootstraps in (**D**) and the error bars in (**E, F**) denote one standard deviation extracted from 100 bootstrap procedures.
The online version of this article includes the following figure supplement(s) for figure 4:

**Figure supplement 1.** SARS-CoV-2 polymerase elongation in presence of T-1106-TP.

---

leading to a 2.6-fold increase in median replication time (*Figure 4B*). For comparison, 50 µM of RDV-TP induced a median replication time as 500 µM T-1106-TP, at the same concentration of competing NTP, suggesting that RDV-TP is better incorporated than T-1106-TP. The final product length was not affected by T-1106-TP, consistent with the T-1106-TP mechanism of action not being terminated (*Figure 4C*; *Dulin et al., 2017*). Performing an experiment using the ultra-stable magnetic tweezers assay (*Bera et al., 2021*) with 500 µM T-1106 and 500 µM all NTPs at 35 pN force, the SARS-CoV-2 polymerase activity traces showed pauses with either a shallow backtrack (*Figure 4—figure supplement 1A*), that is, ≤10 nt, or no significant backtrack at all (*Figure 4—figure supplement 1B*).

Investigating how increasing T-1106-TP concentration affects SARS-CoV-2 RNA synthesis kinetics (*Figure 4D*, *Figure 4—figure supplement 1C*), we found that only the Pause 2 exit rate was

affected, decreasing by tenfold (*Figure 4E*). Pause 1 and Pause 2 probabilities remained constant, while the backtrack pauses increased by almost fivefold, though remaining in the low probability range, that is, ~0.002 (*Figure 4F*). Repeating the experiment at 500 μM NTPs and 300 μM T-1106-TP at 25 pN, we found no difference in comparison to the data acquired at 35 pN (*Figure 2—figure supplement 4E–G*). Here again, the tension has no significant effect. Our results suggest an incorporation of T-1106-TP only via the VSNA pathway (*Figure 1D*), which explains its reduced promiscuity relative to RDV-TP, and is less likely than RDV-TP to induce polymerase backtrack upon incorporation. These observations contrast with our previous findings with poliovirus RdRp (*Dulin et al., 2017*), where T-1106 incorporation induced deep polymerase backtrack. Therefore, the same NA may have a different mechanism of action on different RdRps.

## Sofosbuvir-TP is poorly incorporated by the SARS-CoV-2 polymerase

Next, we compared two uridine analog chain terminators, that is, Sofosbuvir and 3′-dUTP. Sofosbuvir presents a fluoro group at the 2′ α-position and a methyl group at the 2′ β-position, and is a non-obligatory chain terminator. Despite its low incorporation rate (*Villalba et al., 2020*), Sofosbuvir has a proven antiviral effect against hepatitis C virus (HCV) and is an FDA-approved drug to treat HCV infection (*Kayali and Schmidt, 2014*; *Sofia et al., 2010*). It is incorporated by SARS-CoV-2 polymerase (*Gordon et al., 2020b*; *Chien et al., 2020*), but has no efficacy in infected cells (*Xie et al., 2020a*). 3′-dUTP lacks a hydroxyl group in 3′ position, and is therefore an obligatory chain terminator.

The presence of 500 μM Sofosbuvir-TP with 500 μM NTP did not affect RNA synthesis by the SARS-CoV-2 polymerase (*Figure 5A*), while early termination events appeared in the presence of 500 μM 3′-dUTP (*Figure 5B*). Supporting this visual observation, the mean RNA product length of the SARS-CoV-2 polymerase was unaffected by the presence of Sofosbuvir-TP (*Figure 5C*). Raising the 3′-dUTP:UTP stoichiometry to 4 reduced the mean product length by almost fivefold, resulting in an effective incorporation rate $\gamma_{3'-dUTP, 500\,\mu M UTP} = (151 \pm 6)\,\text{nt}$ (*Figure 5D*). For both NAs, the replication time was unaffected (*Figure 5—figure supplement 1A* and *Figure 5—figure supplement 2A*), as well as SARS-CoV-2 RNA synthesis kinetics (*Figure 5—figure supplement 1B–D* and *Figure 5—figure supplement 2B–D*). Reducing the concentration of UTP down to 50 μM while keeping the other NTPs at 500 μM, Sofosbuvir-TP caused few early termination events when increased to 500 μM (*Figure 5E*). Replacing Sofosbuvir-TP by 3′-dUTP, we observed a much stronger effect, as no activity traces reached the end of the template at 3′-dUTP:UTP stoichiometry of 10 (*Figure 5F*). The analysis showed a limited impact of Sofosbuvir-TP on the mean product length, with a minimum of $(563 \pm 32)\,\text{nt}$ at a stoichiometry of 20 (*Figure 5G*). 3′-dUTP was much more effectively incorporated, shortening the mean product length down to $(67 \pm 3)\,\text{nt}$ at the same stoichiometry (*Figure 5H*). Their respective effective incorporation rate at 50 μM UTP reflected these observations, that is, $\gamma_{sofosbuvir, 50\,\mu M UTP} = (3908 \pm 467)\,\text{nt}$ and $\gamma_{3'-dUTP, 50\,\mu M UTP} = (241 \pm 9)\,\text{nt}$. In other words, SARS-CoV-2 polymerase incorporates on average 3908 nt and 241 nt before incorporating either a single Sofosbuvir-TP or a single 3′-dUTP, respectively. The kinetics of RNA synthesis was unaffected by the presence of either 3′-dUTP or Sofosbuvir-TP, while their median replication time decreased at high stoichiometry, a direct consequence of the shortening of the RNA synthesis product (*Figure 5—figure supplement 1E–H* and *Figure 5—figure supplement 2E–H*, respectively). Repeating the experiments for a Sofosbuvir-TP:UTP stoichiometry of 6 now at 25 pN tension (*Figure 2—figure supplement 4A*), we did not see a significant difference in comparison with the data at 35 pN (*Figure 5—figure supplement 1E–H*), therefore the applied tension has no influence in the incorporation of Sofosbuvir-TP. As for 3′-dATP, our data suggest that stoichiometry against the competing NTP regulates Sofosbuvir-TP and 3′-dUTP incorporation, which therefore support that these analogs utilize the NAB state pathway for incorporation (*Figure 1D*). Our data provide further support to the poor incorporation of Sofosbuvir by SARS-CoV-2 (*Gordon et al., 2020b*; *Xie et al., 2020a*) and the low selectivity of the SARS-CoV-2 polymerase against 3′-dUTP.

## ddhCTP is well incorporated by the polymerase but does not affect SARS-CoV-2 replication in cells

3′-Deoxy-3′,4′-didehydro-CTP (ddhCTP) is a recently discovered natural antiviral NA produced in mammalian cells by the viperin-catalyzed conversion of CTP to ddhCTP using a radical-based

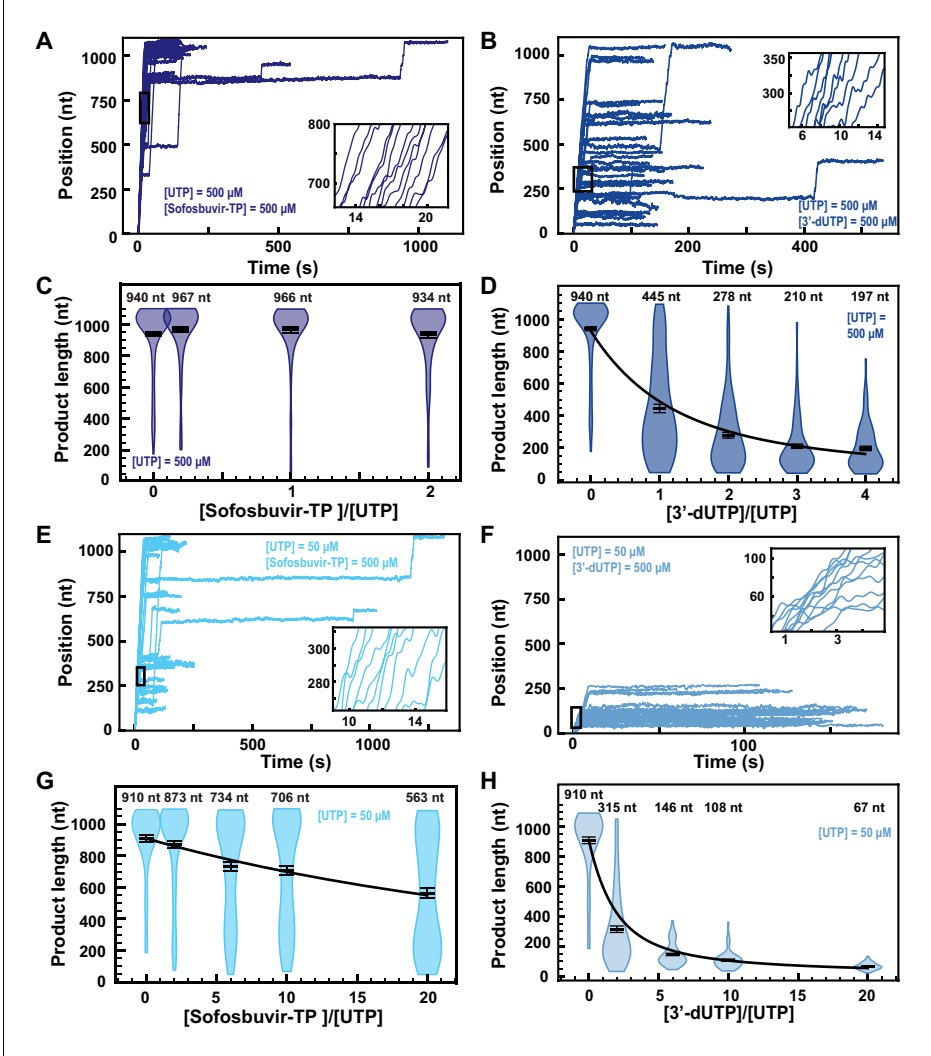

**Figure 5.** Sofosbuvir-TP is a poor SARS-CoV-2 polymerase inhibitor in contrast with 3'-dUTP. (A, B) SARS-CoV-2 polymerase activity traces for 500 µM NTPs and 500 µM of either (A) Sofosbuvir-TP or (B) 3'-dUTP. (C, D) SARS-CoV-2 polymerase product length using the indicated concentration of UTP, 500 µM of other NTPs, as a function of either (C) [Sofosbuvir-TP]/[UTP] or (D) [3'-dUTP]/[UTP]. The mean values are indicated above the violin plots, and represented by horizontal black thick lines flanked by one standard deviation error bars extracted from 1000 bootstraps. (E, F) SARS-CoV-2 polymerase activity traces in the presence of 50 µM of UTP, 500 µM of all other NTPs, and 500 µM of either (E) Sofosbuvir-TP or (F) 3'-dUTP. (G, H) SARS-CoV-2 polymerase product length using 50 µM UTP, 500 µM of other NTPs, as a function of either (G) [Sofosbuvir-TP]/[UTP] or (H) [3'-dUTP]/[UTP]. The mean values are indicated above the violin plots, and represented by horizontal black thick lines flanked by one standard deviation error bars extracted from 1000 bootstraps. In (D, G, H), the solid line is the fit of the terminator effective incorporation rate (see Materials and methods). In (A, B, E, F), the insets are a zoom-in of the replication traces captured in the black square.

The online version of this article includes the following figure supplement(s) for figure 5:

**Figure supplement 1.** SARS-CoV-2 polymerase activity traces kinetics in presence of Sofosbuvir-TP.

**Figure supplement 2.** SARS-CoV-2 polymerase activity traces kinetics in presence of 3'-dUTP.

mechanism (*Gizzi et al., 2018*). While ddhCTP has been shown to efficiently terminate flavivirus replication both in vitro and in cells, its antiviral activity against SARS-CoV-2 polymerase remains unknown. The addition of 500 µM ddhCTP to a reaction buffer containing 500 µM NTP induces early termination events in the SARS-CoV-2 polymerase activity traces (*Figure 6A*). Similar amount of 3'-dCTP instead of ddhCTP resulted in a larger fraction of traces showing early termination events

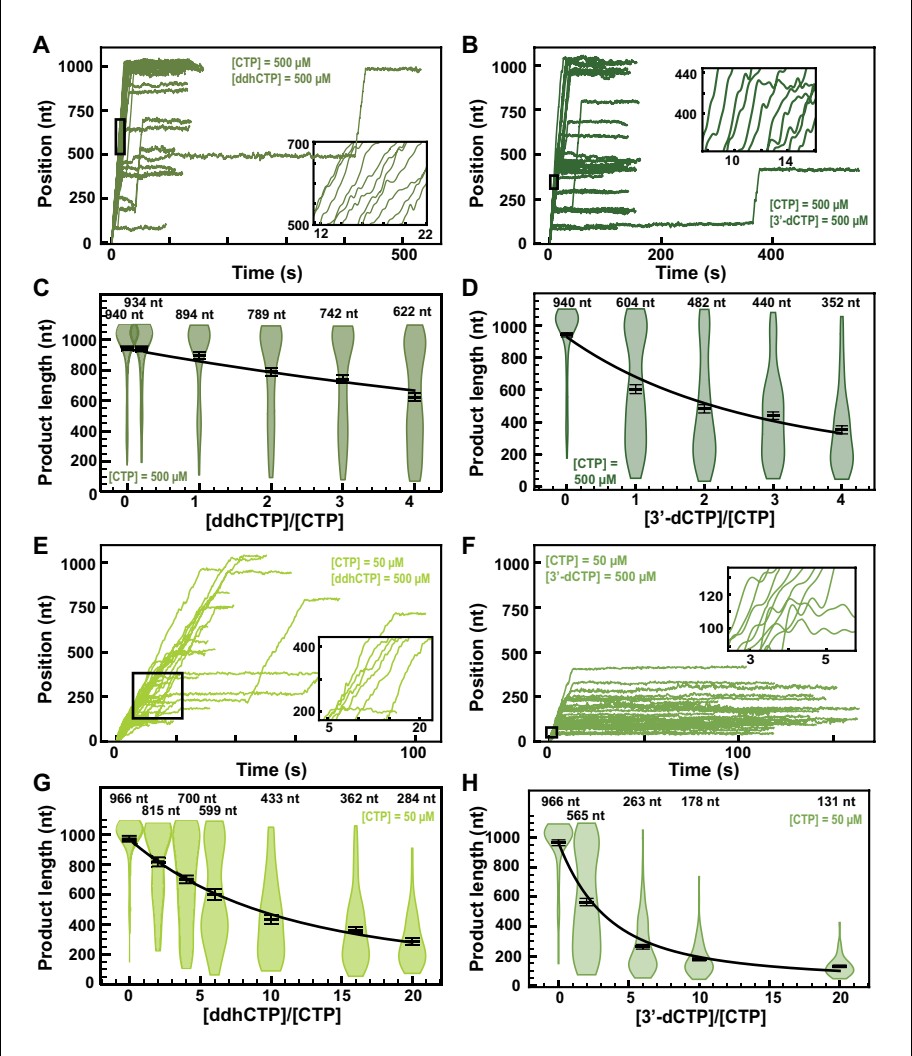

**Figure 6.** ddhCTP and 3'-dCTP inhibit efficiently the SARS-CoV-2 polymerase. (**A, B**) SARS-CoV-2 polymerase activity traces for 500 µM NTPs and 500 µM of either (**A**) ddhCTP or (**B**) 3'-dCTP. (**C, D**) SARS-CoV-2 polymerase product length using the indicated concentration of CTP, 500 µM of other NTPs, as a function of either (**C**) [ddhCTP]/[CTP] or (**D**) [3'-dCTP]/[CTP]. The mean values are indicated above the violin plots, and represented by horizontal black thick lines flanked by one standard deviation error bars extracted from 1000 bootstraps. (**E, F**) SARS-CoV-2 polymerase activity traces in the presence of 50 µM of CTP, 500 µM of all other NTPs, and 500 µM of either (**E**) ddhCTP or (**F**) 3'-dCTP. (**G, H**) SARS-CoV-2 polymerase activity traces product length using 50 µM CTP, 500 µM of other NTPs, as a function of the stoichiometry of either (**G**) [ddhCTP]/[CTP] or (**H**) [3'-dCTP]/[CTP]. The mean values are indicated above the violin plots, and represented by horizontal black thick lines flanked by one standard deviation error bars extracted from 1000 bootstraps. In (**C, D, G, H**), the solid lines are the fits of the terminator effective incorporation rate (see Materials and methods). In (**A, B, E, F**), the insets are a zoom-in of the replication traces captured in the black square.

The online version of this article includes the following source data and figure supplement(s) for figure 6:

**Figure supplement 1.** SARS-CoV-2 polymerase activity traces in presence of ddhCTP.

**Figure supplement 2.** SARS-CoV-2 polymerase activity traces kinetics in presence of 3'-dCTP.

**Figure supplement 3.** ddhC does not inhibit SARS-CoV-2 replication in huh7-hACE2 cells.

**Figure supplement 4.** SARS-CoV-2 nsp14 exoribonuclease knockout is not replicative.

**Figure supplement 4—source data 1.** Source image for the agarose gel in *Figure 6—figure supplement 4D*.

(*Figure 6B*). The average RNA product length of the SARS-CoV-2 polymerase decreased by 1.4-fold when raising the ddhCTP:CTP stoichiometry to 4 (*Figure 6C*), while it decreased by 2.7-fold at similar stoichiometry against CTP (*Figure 6D*). We measured a respective effective incorporation rate $\gamma_{ddhCTP,500\mu\ MCTP} = (1221 \pm 130)\ nt$ and $\gamma_{3'-dCTP,500\mu\ MCTP} = (338 \pm 18)\ nt$ (*Figure 6C,D*). For both NAs, the replication time (*Figure 6—figure supplement 1A* and *Figure 6—figure supplement 2A*) and the RNA synthesis kinetics (*Figure 6—figure supplement 1B–D* and *Figure 6—figure supplement 2B-D*) were largely unaffected. Reducing the concentration of CTP down to 50 µM and keeping the other NTPs at 500 µM, both ddhCTP and 3'-dCTP showed a significant reduction in length of the activity traces (*Figure 6E,F*). Analyzing the average product length, we extracted the respective effective incorporation rates at 50 µM CTP, that is, $\gamma_{ddhCTP,50\mu\ MCTP} = (1360 \pm 71)\ nt$ and $\gamma_{3'-dCTP,50\mu\ MCTP} = (457 \pm 21)\ nt$ (*Figure 6G,H*). These values are similar as what was measured at 500 µM CTP, and confirm the better incorporation of 3'-dCTP over ddhCTP. The kinetics of RNA synthesis was unaffected by the presence of ddhCTP or 3'-dCTP, while their median replication time decreased at high stoichiometry, as a result of the shortening of the RNA synthesis product (*Figure 6—figure supplement 1F–H* and *Figure 6—figure supplement 2F-H*, respectively). We also did not observe any impact of the applied tension for ddhCTP incorporation (*Figure 2—figure supplement 4A*). Here again, stoichiometry against their competing natural nucleotide CTP directly dictates the incorporation of 3'-dCTP and ddhCTP, further supporting the utilization of the NAB pathway for their incorporation (*Figure 1D*).

Though not as high as 3'-dCTP, the effective incorporation rate of ddhCTP should be sufficient to demonstrate a certain efficacy against viral replication in cells. Indeed, ddhCTP is a chain terminator, therefore a single incorporation is sufficient to end RNA synthesis. To verify whether ddhCTP inhibits replication in cells, we infected Huh7-hACE2 cells with SARS-CoV-2, treated these cells with different concentrations of RDV, Sofosbuvir and ddhC, and report on the level of infection by immunofluorescence against SARS-CoV-2 N protein (see Materials and methods, *Figure 6—figure supplement 3A,B*). While RDV showed a clear antiviral effect with an EC$_{50}$ of 0.007 µM (*Figure 6—figure supplement 3B*), ddhC and Sofosbuvir did not show any impact on SARS-CoV-2 replication in cells. This result suggests that SARS-CoV-2 is able to evade the antiviral properties of the endogenously synthesized antiviral NA ddhC. We hypothesized that the 3'–5' exonuclease activity of nsp14 protects SARS-CoV-2 replication by excising ddhCMP from the nascent RNA. To test this hypothesis, we made a SARS-CoV-2 strain, which includes the amino acid substitutions D90A and E92A that remove the exoribonuclease activity of nsp14 (*Figure 6—figure supplement 4A*). This SARS-CoV-2 mutant was unable to replicate in cells (*Figure 6—figure supplement 4B–F*), confirming a recent report (*Ogando et al., 2020*). Therefore, the role of nsp14 in the removal of ddhCMP from the SARS-CoV-2 genome could not be verified experimentally. Future experiments will be designed to address this question.

## Discussion

We present here the first characterization of the mechanism of action of antiviral NAs against SARS-CoV-2 polymerase at the single-molecule level. We show that SARS-CoV-2 polymerase is the fastest RNA studied polymerase to date, elongating up to $\sim 170\ nt.s^{-1}$ at 37 °C (*Figure 1H*). With our assay, we monitored the incorporation and determined the mechanism of action of several NAs, that is, 3'-dATP, 3'-dUTP, 3'-dCTP, Sofosbuvir-TP, ddhCTP, T-1106-TP, and RDV-TP, and resume their properties in *Table 1*.

The present study demonstrates that NA selection and incorporation are not force-dependent (*Figure 2—figure supplement 4*), which further validates the utilization of high-throughput magnetic tweezers to study NA mechanism of action. This result is in agreement with our recent study on SARS-CoV-2 polymerase mechanochemistry, where we showed that entry probability in NAB, SNA, and VSNA was not force-dependent, and that force mainly affected the kinetics of a large conformational subsequent to chemistry, that is, after nucleotide selection and incorporation.

Our study shows that RDV-TP is not a delayed chain terminator at physiological concentration of all NTPs, but instead induces pauses in the polymerase elongation kinetics that are easily overcome at saturating NTP concentration (*Figure 3*). Since our preprint was published on BioRxiv in August 2020, our finding has been corroborated by two recent studies (*Kokic et al., 2021*; *Bravo et al.,*

**Table 1.** Summary table for the investigated NAs.

| | Modification | Incorporation pathway | Mechanism of action | Main conclusions | |
| --- | --- | --- | --- | --- | --- |
| | | | | In vitro incorporation efficiency | In vivo efficacy |
| 3′-dATP | Ribose, 3′ | NAB | Chain terminator | Medium | Unreported |
| 3′-dCTP | Ribose, 3′ | NAB | Chain terminator | Medium | Unreported |
| 3′-dUTP | Ribose, 3′ | NAB | Chain terminator | Medium | Unreported |
| Remdesivir-TP | Ribose, 1′ | SNA, VSNA | Polymerase backtrack | Very high | Very high |
| T-1106-TP | Base | VSNA | Induces pauses (mutagenic) | Medium | Unreported |
| Sofosbuvir-TP | Ribose, 2′ | NAB | Chain terminator | Very low | None |
| ddhCTP | Ribose, 3′ | NAB | Chain terminator | low | None |

*2021*). Similarly, T-1106-TP incorporation does not induce termination, but pauses in the polymerase elongation kinetics (*Figure 4*). However, RDV-TP affects both Pause 1 and Pause 2 exit rates, while T-1106-TP affects only the latter. We showed here that these two NAs do not affect the probability to enter Pause 1 and Pause 2, suggesting that they preferably bind to the polymerase active site after it entered the SNA (Pause 1) or VSNA (Pause 2) pathway. Indeed, if the pauses induced by either RDV-TP or T-1106-TP incorporation were mechanistically unrelated to Pause 1 and Pause 2, the total number of pauses would cumulate and the probability of pausing would dramatically increase, which we do not observe. We therefore suggest that RDV-TP can be incorporated by both SNA and VSNA pathways, while T-1106-TP is only incorporated by the latter. Finally, Pause 1 and Pause 2 respectively account for ~6% and ~0.3% of all the nucleotide addition events at a saturating concentration of NTP. This defines an upper limit for RDV-TP and T-1106-TP relative incorporation, and explains why RDV-TP is incorporated much better than Favipiravir (*Xie et al., 2020a*).

Two recent ensemble kinetic studies investigating the mechanism of action of RDV-TP on SARS-CoV-2 elongation kinetics have recently been published. In the first one, the experiments were performed at submicromolar concentration of NTPs, and showed that RDV-TP is incorporated threefold better than ATP in such conditions (*Gordon et al., 2020b*). In the second one, the authors also claimed that RDV-TP was better incorporated than ATP (*Dangerfield et al., 2020*), while using higher concentration of NTPs than in the first study. Both of these studies agree with our results: RDV is better incorporated by the coronavirus polymerase elongation kinetics at low concentration of natural nucleotides. Indeed, in such conditions, the probabilities of the pathways by which RDV-TP is incorporated, that is, SNA and VSNA, increase significantly (*Bera et al., 2021*). In addition, we showed that RDV-TP incorporation remains noticeable at concentration as low as 20 μM, even when competing with 500 μM ATP. Being able to monitor RDV-TP incorporation at the single-molecule level in competition with saturating concentration of NTP—including ATP—, while the SARS-CoV-2 polymerase was elongating a ~1 kb long RNA product further completes the understanding of RDV mechanism of action.

Our assay revealed that RDV-TP incorporation leads the coronavirus polymerase into backtrack as deep as ~30 nt (*Figure 3GH*). This result demonstrates that the barrier induced by the clash of RDV-MP (*Kokic et al., 2021*) with the serine-861 of nsp12 is sufficiently strong to elicit polymerase backtrack, leading the polymerase into a pause long enough to be mistaken for a termination event in ensemble assays. We anticipate that RDV efficacy is further amplified when the polymerase is elongating through template secondary structures, which stimulates polymerase backtrack (*Bera et al., 2021*). Lower ATP concentration would also decrease the probability to overcome the barrier when an uracil is encoded ~3 nt downstream the incorporated RDV-MP, increasing the backtrack pause probability, as observed here. Interestingly, RDV has a strong efficacy against SARS-CoV-2 in infected cells (*Figure 6—figure supplement 4A,B*), which indicates that the 3′–5′ exonuclease nsp14 does not remove efficiently RDV-MP from the nucleic acid chain. Our results suggest that polymerase backtrack is therefore not an intermediate of product strand proofreading, which corroborates a preceding study showing that nsp14 poorly excise single-stranded RNA (*Ferron et al., 2018*; *Liu et al., 2021*).

Concerning obligatory terminators, the effective incorporation rate we measured showed that 3′-dATP (*Figure 2*), 3′-dUTP (*Figure 5*), 3′-dCTP (*Figure 6*), and—to a lesser extent—ddhCTP (*Figure 6*)

are well incorporated by the SARS-CoV-2 polymerase, while Sofosbuvir-TP is strongly outcompeted by UTP (*Figure 5*). Though well incorporated, 3′-dNTP is cytotoxic, and is therefore not used as antiviral drugs (*Arnold et al., 2012*). Interestingly, the effective incorporation rate of all these terminators is only affected by the stoichiometry of their respective competing natural nucleotide, and not their absolute concentration (unlike RDV-TP), suggesting an incorporation via the NAB pathway (*Figure 1D*). Indeed, we showed that NA incorporated via either the SNA or the VSNA pathway, for example, RDV-TP, would be more likely to be added in the RNA chain at low substrate concentration, independently of the stoichiometry.

A steady-state kinetic study showed that NAs modified at the 2′ and 3′ positions are strongly discriminated against by their competing natural nucleotide (*Gordon et al., 2020b*). Such selectivity is an issue for purine-based analogs, which must compete with high concentrations of ATP and GTP in the cell. In contrast, pyrimidine-based analogs, for example, derivatives of CTP, will only need to compete with intracellular CTP pools on the order of 100 µM (*Traut, 1994*). These features make ddhCTP a particularly attractive antiviral NA. Furthermore, under certain conditions, the interferon α-induced viperin converts up to 30% of the cellular pool of CTP into ddhCTP, further increasing the ddhCTP:CTP stoichiometry in a direction favoring even greater potency (*Gizzi et al., 2018*). However, we could not show any efficacy of ddhC in SARS-CoV-2 infected cells (*Figure 6—figure supplement 4*), suggesting that SARS-CoV-2 has developed ways to counter this cellular defense mechanism. Future studies will investigate whether the exonuclease nsp14 is capable of removing ddhCMP and is therefore responsible for protecting the virus against endogenously produced antiviral NAs.

High-throughput, real-time magnetic tweezers present numerous advantages to study RdRp elongation dynamics, such as monitoring polymerase position with high spatiotemporal resolution while elongating kilobases long templates in the presence of saturating concentration of competing natural nucleotides, and therefore provide complementary information to discontinuous assays to understand the selectivity and/or mechanism of action of NAs. Such an assay will also reveal how adding functional capacity to the core polymerase, for example, RNA helicase activity and proofreading, modulate RdRp elongation dynamics and response to antiviral therapeutics.

## Materials and methods

ddhCTP was prepared as previously described (manuscript in preparation). Briefly, ddhC (*Gizzi et al., 2018*) was dissolved in 20 mM Tris-HCl, 100 mM KCl, and 10 mM BME at pH 7.5. ATP was added to a final concentration of 100 µM, and PEP was added to a concentration of ~3 mM. The proteins human UCK2, CMPK1, and NDK were all added to the reaction mixture to a final concentration of ~10 µM. PK/LDH mixture was added at a final concentration of 1.2 and 1.8 units ml$^{-1}$. After the reaction was complete, proteins were precipitated by lowering the pH to 2 with concentrated HCl and then immediately returning the pH to 9. Precipitated protein was removed by centrifugation and the supernatant was passed through a 0.22 µm filter. The final solution was diluted ten fold using 20 mM TEAB at pH 9.5. ddhCTP was purified with a MonoQ 5/50 anion exchange column using TEAB buffer at pH 9.5. The final ddhCTP was concentrated with lyophilization. Concentration of ddhCTP stocks was determined using an extinction coefficient of 9000 M$^{-1}$ cm$^{-1}$.

### Recombinant protein expression of RdRp (nsp12) and cofactors (nsp7 and nsp8) from SARS-CoV-2

This protocol was described in *Chien et al., 2020*. SARS-CoV-2 nsp12: The SARS-CoV-2 nsp12 gene was codon optimized and cloned into *pFastBac* with C-terminal additions of a TEV site and strep tag (Genscript). The pFastBac plasmid and DH10Bac *Escherichia coli* (Life Technologies) were used to create recombinant bacmids. The bacmid was transfected into Sf9 cells (Expression Systems) with Cellfectin II (Life Technologies) to generate recombinant baculovirus. The baculovirus was amplified through two passages in Sf9 cells, and then used to infect 1 L of Sf21 cells (Expression Systems) and incubated for 48 hr at 27°C. Cells were harvested by centrifugation, resuspended in wash buffer (25 mM HEPES pH 7.4, 300 mM NaCl, 1 mM MgCl$_2$, and 5 mM DTT) with 143 µl of BioLock per liter of culture. Cells were lysed via microfluidization (Microfluidics). Lysates were cleared by centrifugation and filtration. The protein was purified using Strep Tactin superflow agarose (IBA). Strep Tactin eluted protein was further purified by size exclusion chromatography using a Superdex 200 Increase

10/300 column (GE Life Sciences) in 25 mM HEPES, 300 mM NaCl, 100 µM MgCl$_2$, 2 mM TCEP, at pH 7.4. Pure protein was concentrated by ultrafiltration prior to flash freezing in liquid nitrogen. *SARS-CoV-2 nsp7 and nsp8:* The SARS-CoV-2 nsp7 and nsp8 genes were codon optimized and cloned into pET46 (Novagen) with an N-terminal 6× histidine tag, an enterokinase site, and a TEV protease site. Rosetta2 pLys *E. coli* cells (Novagen) were used for bacterial expression. After induction with isopropyl β-D-1-thiogalactopyranoside (IPTG), cultures were grown at 16°C for 16 hr. Cells were harvested by centrifugation and pellets were resuspended in wash buffer (10 mM Tris pH 8.0, 300 mM NaCl, 30 mM imidazole, and 2 mM DTT). Cells were lysed via microfluidization and lysates were cleared by centrifugation and filtration. Proteins were purified using Ni-NTA agarose beads and eluted with wash buffer containing 300 mM imidazole. Eluted nsp12, nsp7, and ns8 were digested with 1% w/w TEV protease during overnight room temperature dialysis (10 mM Tris pH 8.0, 300 mM NaCl, and 2 mM DTT). Digested proteins were passed back over Ni-NTA to remove undigested protein before concentrating the proteins by ultrafiltration. Nsp7 and nsp8 proteins were further purified by size exclusion chromatography using a Superdex 200 Increase 10/300 column (GE Life Sciences). Purified proteins were concentrated by ultrafiltration prior to flash freezing with liquid nitrogen.

## Recombinant protein expression of RdRp (nsp12) and cofactors (nsp7 and nsp8) from SARS-CoV-1

This protocol was described in *Shannon et al., 2020b*. All SARS-CoV proteins used in this study were expressed in *E. coli*, under the control of T5 promoters. Cofactors nsp7L8 and nsp8 alone were expressed from pQE30 vectors with C-terminal and N-terminal hexa-histidine tags, respectively. TEV cleavage site sequences were included for His-tag removal following expression. The nsp7L8 fusion protein was generated by inserting a GSGSGS linker between nsp7- and nsp8-coding sequences. Cofactors were expressed in NEB Express C2523 (New England Biolabs) cells carrying the pRare2-LacI (Novagen) plasmid in the presence of Ampicillin (100 µM/ml) and Chloramphenicol (17 µg/ml). Protein expression was induced with 100 µM IPTG once the OD$_{600}$=0.5–0.6, and expressed overnight at 17°C. Protein was purified first through affinity chromatography with HisPur Cobalt resin (Thermo Fisher Scientific), with a lysis buffer containing 50 mM Tris-HCl pH 8, 300 mM NaCl, 10 mM Imidazole, supplemented with 20 mM MgSO$_4$, 0.25 mg/ml Lysozyme, 10 µg/ml DNase, 1 mM PMSF, with lysis buffer supplemented with 250 mM imidazole. Eluted protein was concentrated and dialyzed overnight in the presence of histidine labeled TEV protease (1:10 w/w ratio to TEV:protein) for removal of the protein tag. Cleaved protein was purified through a second cobalt column and protein was purified through size exclusion chromatography (GE, Superdex S200) in gel filtration buffer (25 mM HEPES pH 8, 150 mM NaCl, 5 mM MgCl$_2$, and 5 mM TCEP). Concentrated aliquots of protein were flash-frozen in liquid nitrogen and stored at −80°C. A synthetic, codon-optimized SARS-CoV nsp12 gene (DNA 2.0) bearing C-terminal 8His-tag preceded by a TEV protease cleavage site was expressed from a pJ404 vector (DNA 2.0) in *E. coli* strain BL21/pG-Tf2 (Takara). Cells were grown at 37°C in the presence of Ampicillin and Chloramphenicol until OD600 reached 2. Cultures were induced with 250 µM IPTG and protein expressed at 17°C overnight. Purification was performed as above in lysis buffer supplemented with 1% CHAPS. Two additional wash steps were performed prior to elution, with buffer supplemented with 20 mM imidazole and 50 mM arginine for the first and second washes respectively. Polymerase was eluted using lysis buffer with 500 mM imidazole and concentrated protein was purified through gel filtration chromatography (GE, Superdex S200) in the same buffer as for nsp7L8. Collected fractions were concentrated and supplemented with 50% glycerol final concentration and stored at −20°C.

## Experimental biosafety while carrying experiments with SARS-CoV-2 infected cells

All experiments involving live SARS-CoV-2 were carried out under biosafety level 3 (BSL-3) containment by personnel wearing the appropriate PPE, including powered air-purifying respirators with Tyvek suits, aprons, booties, and double gloves.

## Cell lines and viruses

Huh7 cells were purchased from Glow Biologics (GBTC-099H) and tested negative for mycoplasma. These cells expressed human ACE2 (huh7-hACE2) after transduction by lentiviral particles derived with pWPI-IRES-Puro-Ak-ACE2 (a gift from Sonja Best; Addgene plasmid # 154985). SARS-CoV-2, isolate USA-WA1/2020 (NR-52281), was obtained through BEI Resources and propagated once on VERO E6 cells before it was used for this study.

## Immunofluorescence assay

Huh7-hACE2 cells in 96-well plates (Corning) were infected with SARS-CoV-2 (USA-WA1/2020 isolate) at MOI of 0.05 in Dulbecco's modified Eagle's medium (DMEM) supplemented with 1% fetal bovine serum (FBS). Before 1.5 hr viral inoculation, the tested compounds were added to the wells in triplicate. The infection proceeded for 24 hr without the removal of the viruses or the compounds. The cells were then fixed with 4% paraformaldehyde, permeabilized with 0.1% Triton-100, blocked with DMEM containing 10% FBS, and stained with a rabbit monoclonal antibody against SARS-CoV-2 NP (GeneTex, GTX635679) and an Alexa Fluor 488-conjugated goat anti-mouse secondary antibody (Thermo Fisher Scientific). Hoechst 33342 was added in the final step to counterstain the nuclei. Fluorescence images of approximately 10,000 cells were acquired per well with a 10× objective in a Cytation 5 (BioTek). The total number of cells, as indicated by the nuclei staining, and the fraction of the infected cells, as indicated by the NP staining, were quantified with the cellular analysis module of the Gen5 software (BioTek).

## SARS-CoV-2 virus production and characterization

SARS-CoV-2 WT and nsp14 exoribonuclease knockout viruses were prepared using a SARS-CoV-2 infectious clone (*Xie et al., 2020b*). Briefly, viral RNA was obtained by in vitro RNA transcription, and 40 μg RNA transcripts and 20 μg N gene RNA were co-electroporated into $8 \times 10^6$ Vero E6 cells using Gene Pulser XCell electroporation system (Bio-Rad, Hercules, CA) at a setting of 270 V and 950 μF with a single pulse. The electroporated cells were seeded to a T75 flask and immediately transfer to BSL-3 facility. Viral production was confirmed by RT-PCR. The supernatants of electroporated cells were harvested and centrifuged at 1000×g for 10 min to remove cell debris. 250 μl supernatant was added and mixed thoroughly with 1 ml of TRIzol LS reagent (Thermo Fisher Scientific). RNA was extracted according to the manufacturer's instructions and resuspended in 20 μl of nuclease-free water. RT-PCR was performed using the SuperScript IV One-Step RT-PCR Kit (Thermo Fisher Scientific).

Virus was determined by plaque assay. Approximately $1.2 \times 10^6$ Vero E6 cells were seeded to each well of a six-well plate. The viruses were tenfold serially diluted with 2% FBS DMEM medium and 200 μl of virus dilution was transferred to each well of the six-well plate. After the incubation for 1 hr at 37℃, 2 ml of overlay medium containing 2% FBS DMEM medium and 1% sea-plaque agarose (Lonza, Walkersville, MD), was added to the infected cells per well. After a 2-day incubation, another 2 ml of overlay medium with neutral red (final concentration 0.01%) was added onto the first overlay. After 12 hr incubation, the plates were sealed with Breath-Easy sealing membrane (Sigma-Aldrich, St. Louis, MO) and plaques were counted.

## SARS-CoV-2 luciferase replicon assay

SARS-CoV-2 transient luciferase replicon assay was performed as previously described (*Xia et al., 2020*). WT and mutant replicon RNA, and N gene mRNA were obtained through T7 in vitro transcription, and 40 μg RNA transcripts and 20 μg N gene RNA were co-electroporated into $8 \times 10^6$ Huh-7 cells (ATCC, tested negative on mycoplasma) using Gene Pulser XCell electroporation system (Bio-Rad) at a setting of 270 V and 950 μF with a single pulse. After 10 min recovery, electroporated cells were seeded to 24-well plates, and harvested at indicated timepoints. Luciferase signal was measured using *Renilla* luciferase assay system (Promega) and read by Cytation 5 (BioTek) according to the manufacturer's protocols.

## Construct fabrication

The fabrication of the RNA hairpin has been described in detail in *Papini et al., 2019*. The RNA hairpin is made of a 499 bp dsRNA stem terminated by a 20 nt loop that is assembled from three ssRNA

annealed together, and two handles, one of 856 bp at the 5′-end and one 822 bp at the 3′-end. The handles include either a 343 nt digoxygenin-labeled ssRNA or a 443 nt biotin-labeled ssRNA. Upon applied force above ~21 pN, the hairpin opens and frees a 1043 nt ssRNA template for SARS-CoV-2 replication. To obtain the different parts of the RNA construct, template DNA fragments were amplified via PCR, purified (Monarch PCR and DNA Cleanup Kit) and in vitro transcribed (NEB HiScribe T7 High Yield RNA Synthesis Kit). Transcripts were then treated with Antarctic Phosphatase and T4 Polynucleotide Kinase. RNAs were purified using the RNA Clean and Concentrator-25 kit (Zymo Research). Individual RNA fragments were annealed and ligated with T4 RNA ligase 2 (NEB) to assemble the RNA hairpin.

The template contains 250 U (24%), 253 A (24%), 273 C (26%), and 267 G (26%).

## High-throughput magnetic tweezers apparatus

The high-throughput magnetic tweezers used in this study have been described in detail elsewhere (*Ostrofet et al., 2018*). Shortly, a pair of vertically aligned permanent magnets (5 mm cubes, Super-Magnete, Switzerland) separated by a 1 mm gap are positioned above a flow cell (see paragraph below) that is mounted on a custom-built inverted microscope. The vertical position and rotation of the magnets are controlled by two linear motors, M-126-PD1 and C-150 (Physik Instrumente PI, GmbH and Co. KG, Karlsruhe, Germany), respectively. The field of view is illuminated through the magnets gap by a collimated LED-light source, and is imaged onto a large chip CMOS camera (Dalsa Falcon2 FA-80–12 M1H, Stemmer Imaging, Germany) using a 50× oil immersion objective (CFI Plan Achro 50 XH, NA 0.9, Nikon, Germany) and an achromatic doublet tube lens of 200 mm focal length and 50 mm diameter (Qioptic, Germany). To control the temperature, we used a system described in detail in *Seifert et al., 2020*. Shortly, a flexible resistive foil heater with an integrated 10 MΩ thermistor (HT10K, Thorlabs) is wrapped around the microscope objective and further insulated by several layers of Kapton tape (KAP22-075, Thorlabs). The heating foil is connected to a PID temperature controller (TC200 PID controller, Thorlabs) to adjust the temperature within ~0.1°.

## Flow cell assembly

The fabrication procedure for flow cells has been described in detail in *Ostrofet et al., 2018*. To summarize, we sandwiched a double layer of Parafilm by two #1 coverslips, the top one having one hole at each end serving as inlet and outlet, the bottom one being coated with a 0.01% m/V nitrocellulose in amyl acetate solution. The flow cell is mounted into a custom-built holder and rinsed with ~1 ml of 1× phosphate-buffered saline (PBS). 3 μm diameter polystyrene reference beads are attached to the bottom coverslip surface by incubating 100 μl of a 1:1000 dilution in PBS of (LB30, Sigma Aldrich, stock conc.: $1.828*10^{11}$ particles per milliliter) for ~3 min. The tethering of the magnetic beads by the RNA hairpin construct relies on a digoxygenin/anti-digoxygenin and biotin-streptavidin attachment at the coverslip surface and the magnetic bead, respectively. Therefore, following a thorough rinsing of the flow cell with PBS, 50 μl of anti-digoxigenin (50 μg/ml in PBS) is incubated for 30 min. The flow cell was flushed with 1 ml of 10 mM Tris, 1 mM EDTA pH 8.0, 750 mM NaCl, 2 mM sodium azide buffer to remove excess of anti-digoxigenin followed by rinsing with another 0.5 ml of 1× TE buffer (10 mM Tris, 1 mM EDTA pH 8.0 supplemented with 150 mM NaCl, and 2 mM sodium azide). The surface is then passivated by incubating bovine serum albumin (BSA, New England Biolabs, 10 mg/ml in PBS and 50% glycerol) for 30 min, and rinsed with 1× TE buffer.

## Single-molecule RdRp replication activity experiments

20 μl of streptavidin-coated Dynal Dynabeads M-270 streptavidin-coated magnetic beads (Thermo Fisher Scientific) was mixed with ~0.1 ng of RNA hairpin (total volume 40 μl) (see Materials and methods) and incubated for ~5 min before rinsing with ~2 ml of 1× TE buffer to remove any unbound RNA and the magnetic beads in excess. RNA tethers were sorted for functional hairpins by looking for the characteristic jump in extension length due to the sudden opening of the hairpin during a force ramp experiment (*Figure 1—figure supplement 1C*; *Papini et al., 2019*). The flow cell was subsequently rinsed with 0.5 ml reaction buffer (50 mM HEPES pH 7.9, 10 mM DTT, 2 μM EDTA, and 5 mM MgCl$_2$). After starting the data acquisition at a force that would keep the hairpin open, 100 μl of reaction buffer containing either 0.6 μM of nsp12, 1.8 μM of nsp7 and nsp8 for SARS-CoV-2 experiments or 0.1 μM of nsp12, 1 μM of nsp7 and nsp8 for SARS-CoV-1 experiments,

the indicated concentration of NTPs and of NAs (if required) were flushed in the flow cell to start the reaction. Sofosbuvir-TP and T-1106-TP were purchased from Jena Bioscience (Jena, Germany) and 3′-dATP was purchased from TriLink Biotechnologies (San Diego, CA). The experiments were conducted at a constant force as indicated for a duration of 20–40 min. The camera frame rate was fixed at either 58 Hz or 200 Hz, for reaction temperature set at either 25°C or 37°C, respectively. A custom written Labview routine (*Cnossen et al., 2014*) controlled the data acquisition and the (x-, y-, z-) positions analysis/tracking of both the magnetic and reference beads in real time. Mechanical drift correction was performed by subtracting the reference bead position to the magnetic bead position.

## Data processing

The replication activity of SARS-CoV-2 core polymerase converts the tether from ssRNA to dsRNA, which concomitantly decreases the end-to-end extension of the tether. The change in extension measured in micron was subsequently converted into replicated nucleotides $N_R$ using the following equation:

$$N_R(F) = N \cdot \frac{L_{ss}(F) - L_{meas}(F)}{L_{ss}(F) - L_{ds}(F)} \tag{1}$$

where $L_{meas}(F)$, $L_{ss}(F)$ and $L_{ds}(F)$ are the measured extension during the experiment, the extension of an ssRNA and of a dsRNA construct, respectively, experiencing a force $F$, and $N$ the number of nucleotides of the ssRNA template (*Dulin et al., 2015a*). The traces were then filtered using a Kaiser-Bessel low-pass filter with a cutoff frequency at 2 Hz. We removed the rare slow outliers traces from data sets (*Figure 1—figure supplement 2A*). As previously described in *Dulin et al., 2015a*, a dwell time analysis was performed by scanning the filtered traces with non-overlapping windows of 10 nt to measure the time (coined throughout the manuscript dwell time) for SARS-CoV-2 polymerase to incorporate ten successive nucleotides. The dwell times of all the traces for a given experimental condition were assembled and further analyzed using a maximum likelihood estimation (MLE) fitting routine to extract the parameters from the stochastic-pausing model.

## SARS-CoV-2 replication product length analysis

To extract the product length of the replication complex, only the traces where the beginning and the end could clearly be distinguished and for which the tether did not rupture for ten minutes following the last observed replication activity were considered. We represented the mean product length, as well as one standard deviation of the mean from 1000 bootstraps as error bars.

## Stochastic-pausing model

The model is described in detail in *Dulin et al., 2017*; *Dulin et al., 2015a*; *Seifert et al., 2020*. There are many kinetic models that are consistent with the empirical dwell time distributions we observe, and we here work under the assumption that the probability of pausing is low enough that there is only one rate-limiting pause in each dwell time window. This assumption washes out most details of the kinetic scheme that connects pauses and nucleotide addition, but allows us to determine the general form of the dwell time distribution without specifying how the pauses are connected to the nucleotide addition pathway

$$p_{\mathrm{dw}}(\mathrm{t}) \propto p_{\mathrm{na}} \Gamma\left(t; N_{\mathrm{dw}}, \frac{1}{k_{\mathrm{na}}}\right) + Q(t) \left(\sum_{n=1}^{N_{\mathrm{sp}}} p_{\mathrm{n}} k n e^{-k_n t} + \frac{a_{bt}}{2(1+\mathrm{t}/1\mathrm{s})^{3/2}}\right) \tag{2}$$

In the above expression, the gamma function in the first term contributes the portion $p_{na}$ of dwell times that originate in the RdRp crossing the dwell time window of size $N_{dw}$ base pairs without pausing; the second term is a sum of contributions originating in pause-dominated transitions, each contributing a fraction $p_n$ of dwell times; the third term captures the asymptotic power-law decay (amplitude $a_{\mathrm{bt}}$) of the probability of dwell times dominated by a backtrack. The backtracked asymptotic term needs to be regularized for times shorter than the diffusive backtrack step. We have introduced a regularization at 1 s, but the precise timescale does not matter, as long as it is set within

the region where the exponential pauses dominate over the backtrack. From left to right, each term of *Equation 1* is dominating the distribution for successively longer dwell times.

A cutoff factor $Q(t)$ for short times is introduced to account for the fact that the dwell time window includes $N_{dw}$ nucleotide-addition steps,

$$Q(t) = \frac{(tk_{na}/N_{dw})^{N_{dw}-1}}{1 + (tk_{na}/N_{dw})^{N_{dw}-1}} \tag{3}$$

The fit results dependence on these cutoffs is negligible as long as they are introduced in regions where the corresponding term is sub-dominant. Here, the cut is placed under the center of the elongation peak, guaranteeing that it is placed where pausing is sub-dominant.

## Maximum likelihood estimation

The normalized version of *Equation 1* is the dwell time distribution fit to the experimentally collected dwell-times $\{t_i\}_i$ by minimizing the likelihood function (*Cowan, 1998*).

$$L = -\sum_i \ln p_{dw}(t_i) \tag{4}$$

with respect to rates and probabilistic weights.

## Dominating in a dwell time window versus dominating in one step

The fractions $p_n$ represent the probability that a particular rate $k_n$ dominates the dwell time. We want to relate this to the probability $P_n$ that a specific exit rate dominates within a 1-nt transcription window. Assuming we have labeled the pauses so that $k_{n-1} > k_n$, we can relate the probability of having rate $n$ dominating in $N_{dw}$ steps to the probability of having it dominate in one step through

$$p_n = \left(\sum_{m=0}^{n} P_m\right)^{N_{dw}} - \left(\sum_{m=0}^{n-1} P_m\right)^{N_{dw}}, p_0 = p_{na} = P_{na}^{N_{dw}} = P_0^{N_{dw}} \tag{5}$$

The first term in *Equation 3* represents the probability of having no pauses longer than the $n^{th}$ pause in the dwell time window, and the second term represents the probability of having no pauses longer than the $(n-1)^{th}$ pause. The difference between the two terms is the probability that the $n^{th}$ pause will dominate. This can be inverted to yield a relation between the single-step probabilities ($P_n$) and the dwell time window probabilities ($p_n$)

$$P_n = \left(\sum_{m=0}^{n} p_m\right)^{1/N_{dw}} - \left(\sum_{m=0}^{n-1} p_m\right)^{1/N_{dw}}, P_0 = p_0^{1/N_{dw}} \tag{6}$$

This relationship has been used throughout the manuscript to relate our fits over a dwell time window to the single-step probabilities.

## Maximum likelihood estimation fitting routine

The above stochastic-pausing model was fit to the dwell time distributions using a custom Python 3.7 routine. Shortly, we implemented a combination of simulated annealing and bound constrained minimization to find the parameters that minimize *Equation 2*. We calculated the statistical error on the parameters by applying the MLE fitting procedure on 100 bootstraps of the original data set (*Press et al., 1992*), and reported the standard deviation for each fitting parameters.

## Competition between obligatory terminator nucleotide analogs and their natural nucleotide homologues

Starting with an empty active site (E), we assume that there is direct binding competition between the natural nucleotide (N) and the NA terminator (T, simply coined terminator) that result in either the former bound (Nb) or the latter bound (Tb) to the active site. From these states there can be any number of intermediate states before the base is either added to the chain with probability $P_{cat}^{T/N}$, or unbinds from the pocket with probability $1 - P_{cat}^{T/N}$ see *Figure 2—figure supplement 3*.

The effective incorporation rate is the attempt rate times the probability of success,

$$k_{\mathrm{inc}}^{\mathrm{T/N}} = [\mathrm{T/N}] K_{\mathrm{on}}^{\mathrm{T/N}} P_{\mathrm{cat}}^{\mathrm{T/N}} \qquad (7)$$

and the relative probability that next incorporated base is a terminator or natural nucleotide is given by the relative effective addition rates.

$$\frac{p^{\mathrm{T}}}{p^{\mathrm{N}}} = \frac{k_{\mathrm{inc}}^{\mathrm{T}}}{k_{\mathrm{inc}}^{\mathrm{N}}} = \frac{[\mathrm{T}]}{[\mathrm{N}]} \frac{K_{\mathrm{on}}^{\mathrm{T}}}{K_{\mathrm{on}}^{\mathrm{N}}} \frac{P_{\mathrm{cat}}^{\mathrm{T}}}{P_{\mathrm{cat}}^{\mathrm{N}}}, p^{\mathrm{T}} + p^{\mathrm{N}} = 1. \qquad (8)$$

This can be rewritten as,

$$p^{\mathrm{N}} = \frac{y}{y+x}, p^{\mathrm{T}} = \frac{x}{y+x}, x = \frac{[\mathrm{T}]}{[\mathrm{N}]}, y = \frac{K_{\mathrm{on}}^{\mathrm{N}} P_{\mathrm{cat}}^{\mathrm{N}}}{K_{\mathrm{on}}^{\mathrm{T}} P_{\mathrm{cat}}^{\mathrm{T}}}$$

In the above, $x$ is the relative stoichiometry between T and N, while $y$ is the relative effective incorporation rates of N and T at equimolar conditions.

On an infinite construct, polymerization will proceed until the first T is incorporated, after which it terminates. At termination, the product has incorporated $n-1$ Ns, and finally one T, with probability.

$$P(n) = \left(p^{\mathrm{N}}\right)^{n-1} p^{\mathrm{T}} = \left(1-p^{\mathrm{T}}\right)^{n-1} p^{\mathrm{T}} \qquad (9)$$

The average number of Ns and Ts incorporated on an infinite construct is therefore.

$$n^{\infty} = \sum_{n=1}^{\infty} n \left(p^{\mathrm{N}}\right)^{n-1} p^{\mathrm{T}} = 1/p^{\mathrm{T}} \qquad (10)$$

If the construct only allows for the addition of $N$ Ns and Ts, the average number of Ns and Ts in the product will instead be,

$$n^N = \sum_{n=1}^{N} n \left(p^{\mathrm{N}}\right)^{n-1} p^{\mathrm{T}} + \sum_{n=N+1}^{\infty} N \left(p^{\mathrm{N}}\right)^{n-1} p^{\mathrm{T}} = \frac{1 - \left(p^{\mathrm{N}}\right)^N}{p^{\mathrm{T}}} = n^{\infty} \left(1 - \left(p^{\mathrm{N}}\right)^N\right) \qquad (11)$$

For a genome of length $L$, with the relative abundance $q$ of templating bases for N and T, we thus expect there to be at most $N = qL$ Ns and Ts incorporated at termination. At termination the product then has the average length.

$$l^L = \frac{n^{qL}}{q} = \frac{1 - \left(p^{\mathrm{N}}\right)^{qL}}{q p^{\mathrm{T}}} = l^{\infty} \left(1 - \left(p^{\mathrm{N}}\right)^{qL}\right), l^{\infty} = \frac{1}{q p^{\mathrm{T}}} \qquad (12)$$

## Data fitting

Though the constructs are 1043 nucleotides long, this length is not always reached even when there are no terminators in the buffer. The average product length is about 10% shorter than the full construct length. To account for this reduction in maximal average product length, we simply fix $L$ to be the mean product length reached without terminator in the buffer, and fit out $\gamma$ from a least-square fit, weighted with the inverse experimental variance.

## Acknowledgements

The authors thank Joy Feng from Gilead Sciences for providing RDV-TP, and Veronique Fattorini and Barbara Selisko for excellent technical assistance and help in SARS-CoV-1 proteins purification. DD was supported by the Interdisciplinary Center for Clinical Research (IZKF) at the University Hospital of the University of Erlangen-Nuremberg, the German Research Foundation grant DFG-DU-1872/3-1 and BaSyC – Building a Synthetic Cell' Gravitation grant (024.003.019) of the Netherlands Ministry of Education, Culture and Science (OCW) and the Netherlands Organisation for Scientific Research (NWO). DD thanks OICE for providing office and lab space, and access to their molecular biology lab. RNK was supported by grant AI123498 from NIAID, NIH. JMW and LDH thank the Ministry of Business Innovation and Employment Contract UOOX1904 (NZ). YX was supported by the

National Institutes of Health (NIH) grant AI151638. SARS-Related Coronavirus 2, Isolate USA-WA1/2020 (NR-52281) was deposited by the Centers for Disease Control and Prevention and obtained through BEI Resources, NIAID, NIH. PYS was supported by NIH grants AI134907 and UL1TR001439, and awards from the Sealy and Smith Foundation, Kleberg Foundation, the John S Dunn Foundation, the Amon G. Carter Foundation, the Gilson Longenbaugh Foundation, and the Summerfield Robert Foundation. JJA and CEC were supported by grant AI045818 from NIAID, NIH. AS, TTNL, and BC acknowledge grants by the Fondation pour la Recherche Médicale (Aide aux équipes), the SCORE project H2020 SC1-PHE-Coronavirus-2020 (grant#101003627), and the REACTing initiative (REsearch and ACTion targeting emerging infectious diseases).

## Additional information

### Funding

| Funder | Grant reference number | Author |
|---|---|---|
| National Institutes of Health | AI123498 | Robert N Kirchdoerfer |
| National Institutes of Health | AI151638 | Yan Xiang |
| National Institutes of Health | AI134907 | Pei-Yong Shi |
| National Institutes of Health | UL1TR001439 | Pei-Yong Shi |
| National Institutes of Health | AI045818 | Jamie J Arnold<br>Craig Eugene Cameron |
| H2020 European Research Council | 101003627 | Bruno Canard |
| Deutsche Forschungsgemeinschaft | DFG-DU-1872/3-1 | David Dulin |
| Nederlandse Organisatie voor Wetenschappelijk Onderzoek | 024.003.019 | David Dulin |
| Ministry of Business, Innovation and Employment | UOOX1904 | James M Wood<br>Lawrence D Harris |

The funders had no role in study design, data collection and interpretation, or the decision to submit the work for publication.

### Author contributions

Mona Seifert, Subhas C Bera, Data curation, Software, Formal analysis, Validation, Investigation, Visualization, Methodology, Writing - review and editing; Pauline van Nies, Resources, Software, Formal analysis, Writing - review and editing; Robert N Kirchdoerfer, Ashleigh Shannon, James M Wood, Lawrence D Harris, Flavia S Papini, Steven Almo, Tyler L Grove, Resources, Writing - review and editing; Thi-Tuyet-Nhung Le, Xiangzhi Meng, Resources, Data curation, Formal analysis, Funding acquisition, Investigation, Visualization, Methodology, Writing - review and editing; Hongjie Xia, Data curation, Formal analysis, Investigation, Visualization, Methodology, Writing - review and editing; Jamie J Arnold, Conceptualization, Resources, Investigation, Writing - review and editing; Pei-Yong Shi, Resources, Formal analysis, Funding acquisition, Visualization, Methodology, Writing - review and editing; Yan Xiang, Data curation, Formal analysis, Funding acquisition, Validation, Investigation, Visualization, Methodology, Writing - review and editing; Bruno Canard, Conceptualization, Resources, Writing - review and editing; Martin Depken, Formal analysis, Visualization, Methodology, Writing - review and editing; Craig E Cameron, Conceptualization, Funding acquisition, Writing - original draft, Writing - review and editing; David Dulin, Conceptualization, Resources, Data curation, Software, Formal analysis, Supervision, Funding acquisition, Validation, Investigation, Visualization, Methodology, Writing - original draft, Project administration, Writing - review and editing

### Author ORCIDs

Mona Seifert ⓘD https://orcid.org/0000-0003-2930-9899
Subhas C Bera ⓘD https://orcid.org/0000-0002-4168-1805

Hongjie Xia [iD] http://orcid.org/0000-0002-2520-7038
James M Wood [iD] https://orcid.org/0000-0002-5965-1006
Lawrence D Harris [iD] https://orcid.org/0000-0002-4214-1018
David Dulin [iD] https://orcid.org/0000-0003-4209-0377

**Decision letter and Author response**
Decision letter https://doi.org/10.7554/eLife.70968.sa1
Author response https://doi.org/10.7554/eLife.70968.sa2

## Additional files

### Supplementary files

• Supplementary file 1. Summary of statistics, rates, and probabilities for each experimental condition presented in this study.

• Supplementary file 2. Summary of statistics, rates, and probabilities for each experimental condition presented in this study.

• Transparent reporting form

### Data availability

We provide two summary tables (Supplementary files 1 and 2) containing all the statistical information and parameter values extracted from the analysis, with their respective error estimates, and in which figures they are represented.

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
