## [Decision Letter]

**Acceptance summary:**

The work presented in this paper may advance our understanding of an important class of anti-viral drugs (nucleoside analogs) that target viral polymerase enzymes by being directly incorporated into the product strand. This class of drugs is known to be quite diverse in their precise mechanisms of action, yet many of the particular details have remained elusive, often due to experimental limitations. The current study employs a single-molecule magnetic-tweezers platform to provide a new paradigm for the mechanism of action of drug remdesivir against the SARS-CoV-2 polymerase. The authors propose that remdesivir does not prevent the complete viral RNA synthesis but causes an increase in polymerase pausing and backtracking. This paper is of broad interest to readers and scientists working on SARS-CoV-2 and other RNA-based viruses. It will be also of interest to researchers studying RNA polymerase mechanisms and evolution.

**Decision letter after peer review:**

Thank you for submitting your article "Inhibition of SARS-CoV-2 polymerase by nucleotide analogs: a single molecule perspective" for consideration by *eLife*. Your article has been reviewed by 3 peer reviewers, and the evaluation has been overseen by Maria Spies as a Reviewing Editor and Aleksandra Walczak as the Senior Editor. The reviewers have opted to remain anonymous.

All three reviewers and the reviewing editor agreed that the study is interesting, technically sound and has merit. All, however, thought that there are several overstatements that need to be tempered (below).

1) The practicality of the magnetic tweezer assay as a tool for drug discovery is overstated. Please change the verbiage.

2) Explain the significance and the mechanism of observed backtracking in the absence of the nuclease domain.

3) How the obtained results translate into more physiological conditions at zero force should be addressed.

4) The paper would benefit of including a table summarizing the effect of the different NA in the polymerase activity, and pointing out the main conclusions for each compound.

5) Address the reviewers' individual comments.

*Reviewer #1:*

This work investigated the mechanism of inhibition of SARS-CoV-2 polymerase by multiple nucleotide analogs using a high-throughput, single-molecule, magnetic tweezers platform. There was particular focus on the remdesivir (RDV) because it is the only FDA approved anti-coronavirus drug on the market at the time of this review. The study shows that remdesivir leads the polymerase to undergo a backtrack in which it moves back as much as 30 nucleotides from the last insertion. The results also show that RDV is not a chain terminator, which is consistent with prior work. In addition to RDV, the authors characterized other nucleotide analogs such as ddhCTP, 3'-dCTP, and Sofosbuvir-TP to propose that the location of the modification in the ribose or in the base dictates the catalytic pathway used for incorporation. The authors also propose that the use of magnetic tweezers is essential towards characterizing and discovering therapeutics that target viral polymerases.

Strengths:

A strength of the papers is the utilization of magnetic tweezers to characterize the polymerase at the single molecule level. This provides a unique method to capture less common or difficult to observe phenomena such as backtracking. Most bulk ensemble assays would have difficulty detecting these phenomena.

The characterization of multiple different types of nucleotides analogs to investigate the different mechanisms by which they could inhibit the polymerase is a strength of the paper. The authors elegantly utilize their system to show different pause states and backtracking of the polymerase.

In general, the paper is well written, and the data is clearly presented.

Weakness:

The experiments performed with the magnetic tweezers appear to not have contained the exonuclease domain. This domain would presumably be involved in removing nucleotide analogs that have been inserted and may alter the pause states or backtracking prevalence. For example, does the prevalence of backtracking increase when the exonuclease domain is not present. This is particularly important in regard to the RDV experiments.

A major claim for this study is the utilization of the magnetic tweezers "experimental paradigm" as being essential to the discovery and development of therapeutics to viral polymerases. In addition the authors state this approach is superior to bulk ensemble studies. This reviewer found these conclusions to be an overstatement and unnecessary. The use of magnetic tweezers is not amenable to all laboratories or an easy technique to implement within the therapeutic drug development. In general, the authors also overstate the power and feasibility of the magnetic tweezers in comparison to bulk ensemble studies. All assays have limitations, and the magnetic tweezers is no different in regards to being purified proteins, an in vitro approach, limitations in regards to feasibility for all users, ability to detect the amount of active protein, and multiple other reasons. This is a minor weakness of the paper that can be easily addressed because it detracts from the novelty of the studies.

1) The term backtracked is a bit confusing to this reviewer and likely will be to many other readers. When the polymerase backtracks does it remove the inserted nucleotides or just slide backwards. This should be clearly defined in this manuscript and a figure may help.

2) If the polymerase is back tracking within the cell, does it invoke the exonuclease activity to remove the nucleotide analog and continue extension? It appears the experiments were performed with a polymerase lacking the exonuclease domain. This raises questions if the extensive backtracking with RDV is a result of the missing exonuclease domain. Can the same experiment be performed with the exonuclease proficient protein to verify the general trend of deep backtracking by RDV?

3) What is the active fraction of enzyme used in these assays? This is an important control to ensure the protein preps contain a high percentage of active enzyme and does not contain a subpopulation of inactive enzyme. Especially given reports of His-tagged protein resulting in lower activity (see ref 21, PMID: 33283177), which was the purification scheme used in this study.

4) There are a number of presentation issues with the manuscript that distract from the scientific results presented, which are interesting. At times it often felt this was to enhance the perceived impact of the paper or method used. I have listed a few below and would suggest the authors alter the tone.

a. The authors state their study definitely proves the RDV-TP is not a chain terminator in the discussion and results. This implies that this is novel or that other groups have reported RDV-TP to be a chain terminator(?). However, in the very next sentence the authors point out that RDV-TP was recently reported to not be a chain terminator in corroborating studies (ref 29,39). This should be clarified for the reader.

b. In the last paragraph of the discussion the authors state, "High-throughput, real-time magnetic tweezers present numerous advantages to study RdRp elongation dynamics over a discontinuous assay, and therefore demand integration of such an assay into any pipeline investigating the selectivity and/or mechanism of action of NAs." This statement is not accurate in terms of "demanding integration" and the "numerous advantages over a discontinuous assay" is subjective. I would suggest removing or altering the tone of these statements. The approach is powerful and provides mechanistic insight but doesn't demand integration. Furthermore, each assay has benefits that another type of assay may lack. The commentary about one assay being better than another does not strengthen the paper.

c. In the discussion the authors have a paragraph dedicated to two ensemble studies that showed that RDV-TP is better inserted than ATP (ref 17 and 21). The authors generally state these studies did not mimic in vivo conditions or utilize saturating concentration of ATP. This seems to all be in regard to claiming that their magnetic tweezers approach is superior to the ensemble. However, the conclusions from both of the prior ensemble studies in regard to RDV-TP being better inserted than ATP appear to be sound, and this was not addressed in this study, nor can it be (easily) with the magnetic tweezers approach. Therefore, the conclusions from the ensemble studies are accurate in this reviewer's opinion and the paragraph in the discussion should be removed/altered. In addition, the Johnson lab doesn't need a competing concentration of ATP to determine the kinetic values and thus incorporation rate based on the design of the experiments. It also is not clear what the authors mean by the ensemble studies are performed at subsaturating concentrations of substrate. These experiments are purposely designed to be done this way and the nucleotide concentration was at a saturating concentration of the pol/DNA complex which was rate limited by the DNA concentration. Finally, neither the ensemble or magnetic tweezers approaches are representative of in vivo conditions as they are both in vitro with their own limitations.

*Reviewer #2:*

This study investigates the impact of remdesivir (RDV) and other nucleotide analogs (NAs), 3'-dATP, 3'-dUTP, 3'-dCTP, Sofosbuvir-TP, ddhCTP, and T-1106-TP, on RNA synthesis by the SARS-CoV-2 polymerase using magnetic tweezer. This technique allows to directly quantify termination of viral synthesis, pausing or stalling of the polymerase, thus, defining the effect of these NAs on viral synthesis. The work includes good quality data and nicely stablishes an assay to follow the activity of the SARS-CoV-2 RNA-dependent RNA polymerase. However, the basis of the assay and theory was largely presented before by the authors in Ref 22 and 23 (and other references therein). The main result here is that RDV incorporation does not prevent the complete viral RNA synthesis but causes an increase of pausing and back-tracking. This contrasts with a clear signature of synthesis termination induced by 3'-dATP. The work is complemented with the characterization of other NAs. Despite these results are of merit, I do not see this work to present a sufficient advance of our current knowledge. How these results translate into more physiological conditions at zero force should be addressed. The rationale of testing other NAs apart from the mere systematic characterization of other compounds is unclear. Similarly, I do not see the benefits of adding cell experiments with three compounds and experiments with the nsp14 mutant to address proofreading because they were inconclusive.

1) Abstract. The term "deep backtrack" appearing in the abstract is unclear. Moreover, I would also avoid the use of word "unambiguously" in the abstract. I guess a claim on something cannot be ambiguous.

2) The force at which MT experiments were done is important. Although mentioned in the Methods, the force should be also indicated in Figure 1A, and in the main text. This will help understand the assay and movement of the bead upon nucleotide incorporation. The mechanical characterization of the initial (ssRNA) and final (dsRNA) substrates should be included as supplementary.

3) Experiments were done at 35 and 25 pN. These are far from physiological forces, likely to be near zero. How would this might affect the reported conclusions? It would be good to complement with some (bulk?) experiments done at zero or near zero force, to backup for instance some of the claims on product length and termination. Additionally, I expect the equivalent MT experiment at low force to result in the increase of bead vertical position due to the conversion of ssRNA to dsRNA. Did the authors try that?

4) Related to previous point, the fact that results were similar at 25 and 35 pN is not sufficient to claim that "tension does not play a significant role in RDV-TP incorporation". This claim is also repeated for other nucleotides analogues. Authors tested very particular conditions and such general conclusion cannot be extracted from that data.

5) Protein purification. Biochemical characterization of the purified proteins should be included, for instance a SDS-PAGE with purified proteins.

6) Figure 2 —figure supplement 1 is cited before than Figure 2. Not very logical. Suggest perhaps including the structure of each compounds in the main figure where used?

7) Authors claim that their assay works at saturating NTP concentration, an issue they believe is problematic with other published works (Refs 17 and 21). Could the authors determine the ratio of incorporation of RDV-TP versus ATP to contrast with the other published works (Refs 17, 21)?

8) Authors claim incorporation of RDV-TP leads to backtracking, but not to proofreading. This is a bit confusing. What would it be the function of backtracking here?

9) The paper would benefit of including a table summarizing the effect of the different NA in the polymerase activity, and pointing out the main conclusions for each compound.

10) The description of the model and the analysis of dwell time distributions is too technical and difficult to follow in its current form. Equations are not numbered, terminology not defined, a figure is embedded within the text… I find all these developments more appropriate for a more specialized journal.

11) I guess the experiments using cells infected with SARS-CoV2 required a special biohazard security lab and specific measures. Please indicate all these details.

12) I do not see the benefit of including these cell experiments (Figure 6 – S3-S4), in their present form. Perhaps, a proper motivation is missing. The experiments with the nsp14 mutant are not conclusive and do not see the reason to include them.

*Reviewer #3:*

This manuscript focuses on understanding the mechanism of action of remdesivir in the inhibition of SARS-Cov2 polymerase, using single molecule methods. The findings are highly original, significant and surprising. The approach is highly robust and supported by a range of orthogonal studies. Overall, these findings should help those engaged directly in drug discovery by providing a critical foundational understanding for the action of remdesivir.

The research described in this manuscript has several findings that significantly impact the broader field polymerase inhibition. First, the authors were able to show using single molecule methods that remdesivir-TP incorporation leads to polymerase backtrack. This is important because the pause is long enough that an ensemble assay could mistake this backtrack for a termination event. Secondly, the researchers found the effective incorporation of remdesivir-TP was determined by its absolute concentration. This suggests remdesivir-TP and similar nucleotide analogs incorporate via the SNA or VSNA pathway and would be more likely to add to the RNA chain when substrate concentration is low (independent of stoichiometry with the competing native nucleotide). Thirdly, the researchers found the effective incorporation rate of obligatory terminators was affected by the stoichiometry of their competing native nucleotide rather than their absolute concentration. This suggests that obligatory terminators are incorporated via the NAB pathway. The pausing that the researchers observed in the polymerase elongation kinetics have recently been demonstrated by two other groups. However, this study improved upon the assay conditions used by other researchers to recapitulate in vivo conditions and remove bias from kinetics measurements.

The authors highlighted the issues with remdesivir, tested other nucleotide analogs, and proposed a better alternative based on their assays (ddhCTP). Interestingly, the ddhCTP didn't actually work in infected cells. However, the authors presented a few theories on why it didn't work and said they plan to follow up to elucidate why it didn't work in cells. I think those results will be very interesting for the larger community working in this area. It's clear that the authors made a substantial enough contribution on the mechanism of inhibition of SARS Cov2 polymerase to merit publication in *eLife*, independent of the work on the "improved" antiviral candidate.

It would have been useful to clarify for the reader the pharmaceutical import of the putative delayed chain termination (or pausing) relative to actual chemical chain termination. In other words, I'm assuming that in both cases the viral genome is considered to be non-transcribed (in that a chemical agent has been incorporated into the growing strand). This is true for most compounds in this broad class of anti-virals. The issues are usually surrounding the width of the therapeutic index and the degree to which resistant mutants arise.

Overall, this manuscript constitutes a major advance in our understanding of chain termination in polymerases, and provides deep insights into the mechanism of action of remdesivir, which may contribute to further drug discovery efforts targeting this polymerase. Additionally, the authors have highlighted and addressed issues in the methodologies of previous mechanistic studies that led others to erroneous conclusions.

---

## [Author Response]

Reviewer #1:This work investigated the mechanism of inhibition of SARS-CoV-2 polymerase by multiple nucleotide analogs using a high-throughput, single-molecule, magnetic tweezers platform. There was particular focus on the remdesivir (RDV) because it is the only FDA approved anti-coronavirus drug on the market at the time of this review. The study shows that remdesivir leads the polymerase to undergo a backtrack in which it moves back as much as 30 nucleotides from the last insertion. The results also show that RDV is not a chain terminator, which is consistent with prior work. In addition to RDV, the authors characterized other nucleotide analogs such as ddhCTP, 3'-dCTP, and Sofosbuvir-TP to propose that the location of the modification in the ribose or in the base dictates the catalytic pathway used for incorporation. The authors also propose that the use of magnetic tweezers is essential towards characterizing and discovering therapeutics that target viral polymerases.Strengths:A strength of the papers is the utilization of magnetic tweezers to characterize the polymerase at the single molecule level. This provides a unique method to capture less common or difficult to observe phenomena such as backtracking. Most bulk ensemble assays would have difficulty detecting these phenomena.The characterization of multiple different types of nucleotides analogs to investigate the different mechanisms by which they could inhibit the polymerase is a strength of the paper. The authors elegantly utilize their system to show different pause states and backtracking of the polymerase.In general, the paper is well written, and the data is clearly presented.

The authors thank the Reviewer for the strong appraisal of our work.

Weakness:The experiments performed with the magnetic tweezers appear to not have contained the exonuclease domain. This domain would presumably be involved in removing nucleotide analogs that have been inserted and may alter the pause states or backtracking prevalence. For example, does the prevalence of backtracking increase when the exonuclease domain is not present. This is particularly important in regard to the RDV experiments.

To date, no laboratory has been able to couple the polymerase complex with the proofreading complex. Indeed, we have entire five-year R01 grant to pursue this objective. Just like all proofreading polymerases studied before this one, it is imperative to establish a baseline with exonuclease deficient state prior to adding that component. Even before we add the exonuclease, it will be important to add the helicase to determine if it can assist the polymerase with dsRNA, because its strand-displacement activity is weak.

A major claim for this study is the utilization of the magnetic tweezers "experimental paradigm" as being essential to the discovery and development of therapeutics to viral polymerases. In addition the authors state this approach is superior to bulk ensemble studies. This reviewer found these conclusions to be an overstatement and unnecessary. The use of magnetic tweezers is not amenable to all laboratories or an easy technique to implement within the therapeutic drug development. In general, the authors also overstate the power and feasibility of the magnetic tweezers in comparison to bulk ensemble studies. All assays have limitations, and the magnetic tweezers is no different in regards to being purified proteins, an in vitro approach, limitations in regards to feasibility for all users, ability to detect the amount of active protein, and multiple other reasons. This is a minor weakness of the paper that can be easily addressed because it detracts from the novelty of the studies.

We feel that it is important to avoid an either-or scenario. We apologize for evoking a negative reaction with our statement, as we were only trying to emphasize how illuminating the magnetic-tweezers approach can be. It was not our intention to rule out the need for bulk methods at the bench top or using quench-flow or stopped-flow devices.

We have edited the text in l.83-87 to convey the following:

“Magnetic tweezers permit the dynamics of an elongating polymerase/polymerase complex to be monitored in real time and the impact of nucleotide analogues to be monitored in the presence of all four natural nucleotides in their physiological concentration ranges. Here, we present a magnetic tweezers assay to provide insights into the mechanism and efficacy of current and underexplored NAs on the coronavirus polymerase.”

1) The term backtracked is a bit confusing to this reviewer and likely will be to many other readers. When the polymerase backtracks does it remove the inserted nucleotides or just slide backwards. This should be clearly defined in this manuscript and a figure may help.

As ‘Backtracking’ is a term commonly used in the field of RNA polymerase, and that we have introduced it in our single molecule investigations of several viral RdRp’s (Dulin et al., Cell Reports 2015; 2017; NAR 2015; Seifert et al., NAR 2020), including SARS-CoV-2 polymerase (Bera et al., BioRxiv 2021), which was also structurally characterized (Malone et al., PNAS 2021), we did not think it would be confusing. We have added a schematic in Figure 3G and amended the text with in l.125 to clarify this concept:

“while the long-lived pauses relate to a catalytically incompetent polymerase backtrack state, where the polymerase diffuses backward on the template strand, leading the product strand 3’-end to unwind and exit via the NTP channel without cleavage.”

2) If the polymerase is back tracking within the cell, does it invoke the exonuclease activity to remove the nucleotide analog and continue extension? It appears the experiments were performed with a polymerase lacking the exonuclease domain. This raises questions if the extensive backtracking with RDV is a result of the missing exonuclease domain. Can the same experiment be performed with the exonuclease proficient protein to verify the general trend of deep backtracking by RDV?

The connection from backtrack to proofreading was done in a structural paper (Malone et al., PNAS 2021) as an analogy to the cellular multi-subunits DNA-dependent RNA polymerases, to which the coronavirus polymerase is evolutionary and structurally not related. Backtracking is not part of the proofreading process for A-family polymerases, to which the coronavirus polymerase is related. Therefore, we do not make the connection between backtracking and proofreading, as there is neither functional nor structural evidence for this function. Nsp14 is not a domain of the polymerase, but an enzyme by itself that putatively associates to the polymerase complex. To date, there has been no publication with a functional assay combining the polymerase and the exonuclease (nsp14) proving the two enzymes were acting as a complex. Furthermore, there isn’t a clear understanding on the nature and structure of a polymerase complex including nsp14 (and maybe nsp10). The only activity observed was on separate enzymes, showing that nsp14 efficiently excises 3’-end RNA strand in a dsRNA configuration, much less efficiently on a single stranded RNA (Liu et al., Science 2021), as it would be the case for a backtracked RNA. A single molecule assay studying the coronavirus polymerase in complex with nsp14 would be a study by itself and extend beyond the scope of the present work.

3) What is the active fraction of enzyme used in these assays? This is an important control to ensure the protein preps contain a high percentage of active enzyme and does not contain a subpopulation of inactive enzyme. Especially given reports of His-tagged protein resulting in lower activity (see ref 21, PMID: 33283177), which was the purification scheme used in this study.

In the Materials and methods (l. 540-546), we replaced “Eluted proteins” with “Eluted nsp12, nsp7 and nsp8”, which now reads as:

“Eluted nsp12, nsp7 and nsp8 were digested with 1% w/w TEV protease during overnight room temperature dialysis (10 mM Tris pH 8.0, 300 mM NaCl, 2 mM DTT). Digested proteins were passed back over Ni-NTA to remove undigested protein before concentrating the proteins by ultrafiltration. Nsp7 and nsp8 proteins were further purified by size exclusion chromatography using a Superdex 200 Increase 10/300 column (GE Life Sciences). Purified proteins were concentrated by ultrafiltration prior to flash freezing with liquid nitrogen.”

Our assay demonstrates ~80% activity in the conditions described in the Materials and methods. This means 80% of all the tethers in the field of view (hundreds, see Figure 1—figure supplement 1D) show efficient and consistent primer elongation activity, and support that 80% of the complex that forms at the primer template are elongation competent. We also showed in the companion article (attached with the present submission, https://doi.org/10.1101/2021.03.27.437309) that we were able to form stable and processive elongation complex.

Furthermore, the added benefit of a single-molecule approach is that only functional complexes are observed, so the fraction of “active” enzyme is not as important for investigations using this method as it would be for bulk assays.

4) There are a number of presentation issues with the manuscript that distract from the scientific results presented, which are interesting. At times it often felt this was to enhance the perceived impact of the paper or method used. I have listed a few below and would suggest the authors alter the tone.a. The authors state their study definitely proves the RDV-TP is not a chain terminator in the discussion and results. This implies that this is novel or that other groups have reported RDV-TP to be a chain terminator(?). However, in the very next sentence the authors point out that RDV-TP was recently reported to not be a chain terminator in corroborating studies (ref 29,39). This should be clarified for the reader.

We take this opportunity to mention again that our study was the first to report that RDV-TP is neither a chain terminator nor a delayed chain terminator. We have amended the text in l.416-419 for clarity such as:

“Our study shows that RDV-TP is not a delayed chain terminator at physiological concentration of all NTPs, but instead induces pauses in the polymerase elongation kinetics that are easily overcome at saturating NTP concentration (Figure 3). Since our preprint was published on BioRxiv in August 2020, our finding has been corroborated by two recent studies”

b. In the last paragraph of the discussion the authors state, "High-throughput, real-time magnetic tweezers present numerous advantages to study RdRp elongation dynamics over a discontinuous assay, and therefore demand integration of such an assay into any pipeline investigating the selectivity and/or mechanism of action of NAs." This statement is not accurate in terms of "demanding integration" and the "numerous advantages over a discontinuous assay" is subjective. I would suggest removing or altering the tone of these statements. The approach is powerful and provides mechanistic insight but doesn't demand integration. Furthermore, each assay has benefits that another type of assay may lack. The commentary about one assay being better than another does not strengthen the paper.

Reviewer #1 is right that single-molecule high-throughput magnetic tweezers and discontinuous assays present different advantages, and we did not mean to say our approach is better than discontinuous assays. For example, discontinuous assays are the best approach to characterize template sequence dependence kinetics. We agree that our writing was not specific enough and may be misleading. We have amended the introduction such as:

“Magnetic tweezers permit the dynamics of an elongating polymerase/polymerase complex to be monitored in real time and the impact of nucleotide analogues to be monitored in the presence of all four natural nucleotides in their physiological concentration ranges. Here, we present a magnetic tweezers assay to provide insights into the mechanism and efficacy of current and underexplored NAs on the coronavirus polymerase.”

And we have modified the discussion in l.518-521 to be more specific on the advantages of magnetic tweezers:

“High-throughput, real-time magnetic tweezers present numerous advantages to study RdRp elongation dynamics, such as monitoring polymerase position with high spatiotemporal resolution while elongating kilobases long templates in the presence of saturating concentration of competing natural nucleotides, and therefore provide complementary information to discontinuous assays to understand the selectivity and/or mechanism of action of NAs.”

c. In the discussion the authors have a paragraph dedicated to two ensemble studies that showed that RDV-TP is better inserted than ATP (ref 17 and 21). The authors generally state these studies did not mimic in vivo conditions or utilize saturating concentration of ATP. This seems to all be in regard to claiming that their magnetic tweezers approach is superior to the ensemble. However, the conclusions from both of the prior ensemble studies in regard to RDV-TP being better inserted than ATP appear to be sound, and this was not addressed in this study, nor can it be (easily) with the magnetic tweezers approach. Therefore, the conclusions from the ensemble studies are accurate in this reviewer's opinion and the paragraph in the discussion should be removed/altered. In addition, the Johnson lab doesn't need a competing concentration of ATP to determine the kinetic values and thus incorporation rate based on the design of the experiments. It also is not clear what the authors mean by the ensemble studies are performed at subsaturating concentrations of substrate. These experiments are purposely designed to be done this way and the nucleotide concentration was at a saturating concentration of the pol/DNA complex which was rate limited by the DNA concentration. Finally, neither the ensemble or magnetic tweezers approaches are representative of in vivo conditions as they are both in vitro with their own limitations.

Here again, it was not our intention to say that magnetic tweezers are better than ensemble approaches. We believe these are complementary approaches. We have amended the discussion paragraph in l.461-474, which now reads:

“Two recent ensemble kinetic studies investigating the mechanism of action of RDV-TP on SARS-CoV-2 elongation kinetics have recently been published. In the first one, the experiments were performed at submicromolar concentration of NTPs, and showed that RDV-TP is incorporated 3-fold better than ATP in such condition. In the second one, the authors also claimed that RDV-TP was better incorporated than ATP, while using higher concentration of NTPs than in the first study. Both of these studies agree with our results: Remdesivir is better incorporated by the coronavirus polymerase elongation kinetics at low concentration of natural nucleotides. Indeed, in such condition, the probabilities of the pathways by which RDV-TP is incorporated, i.e. SNA and VSNA, increase significantly. In addition, we showed that RDV-TP incorporation remain noticeable at concentration as low as 20 µM, even when competing with 500 µM ATP. Being able to monitor RDV-TP incorporation at the single molecule level in competition with saturating concentration of NTP – including ATP –, while the SARS-CoV-2 polymerase was elongating a ~1 kb long RNA product further completes the understanding of RDV mechanism of action”

Reviewer #2:This study investigates the impact of remdesivir (RDV) and other nucleotide analogs (NAs), 3'-dATP, 3'-dUTP, 3'-dCTP, Sofosbuvir-TP, ddhCTP, and T-1106-TP, on RNA synthesis by the SARS-CoV-2 polymerase using magnetic tweezer. This technique allows to directly quantify termination of viral synthesis, pausing or stalling of the polymerase, thus, defining the effect of these NAs on viral synthesis. The work includes good quality data and nicely stablishes an assay to follow the activity of the SARS-CoV-2 RNA-dependent RNA polymerase.

The authors thank the Reviewer for her/his appreciation of our work.

However, the basis of the assay and theory was largely presented before by the authors in Ref 22 and 23 (and other references therein). The main result here is that RDV incorporation does not prevent the complete viral RNA synthesis but causes an increase of pausing and back-tracking. This contrasts with a clear signature of synthesis termination induced by 3'-dATP. The work is complemented with the characterization of other NAs. Despite these results are of merit, I do not see this work to present a sufficient advance of our current knowledge.

We acknowledge Reviewer #2 opinion. However, we believe that our work is highly novel and important, as noted by Reviewer #1: “This [utilization of magnetic tweezers] provides a unique method to capture less common or difficult to observe phenomena such as backtracking. Most bulk ensemble assays would have difficulty detecting these phenomena.”

and Reviewer #3: “Overall, this manuscript constitutes a major advance in our understanding of chain termination in polymerases, and provides deep insights into the mechanism of action of remdesivir, which may contribute to further drug discovery efforts targeting this polymerase.”.

How these results translate into more physiological conditions at zero force should be addressed.

We show here that nucleotide analogs are incorporated via specific catalytic pathways (NAB, SNA, VSNA) depending on the nature of their modification (position and type in ribose, base). In the companion paper attached to this submission (https://doi.org/10.1101/2021.03.27.437309, currently in press), we show that the force has no effect on the probability to enter any catalytic pathways, and only affects the kinetics of a large conformational change occurring after chemistry. In conclusion, the force has no effect on nucleotide analog selection, as supported by our evaluation at both 25 and 35 pN. To clarify this, we have added in l.416-421:

“The present study demonstrates that nucleotide analog selection and incorporation is not force-dependent (Figure 2—figure supplement 3), which further validates the utilization of high-throughput magnetic tweezers to study nucleotide analog mechanism of action. This result is in agreement with our recent SARS-CoV-2 polymerase mechanochemistry paper, where we showed that entry probability in NAB, SNA and VSNA was not force dependent, and that force mainly affected the kinetics of a large conformational subsequent to chemistry, i.e. after nucleotide selection and incorporation.”

The rationale of testing other NAs apart from the mere systematic characterization of other compounds is unclear.

We have tested 3’-dATP, a well-known chain terminator, with Remdesivir, which was claimed to be a delayed chain terminator, as both are ATP analogue. We monitored the incorporation of Sofosbuvir, a well-known inhibitor of HCV replication, with its 3’-dNTP homologue, i.e. 3’-dUTP. T-1106-TP is a compound that was recently tested for coronavirus because it has a proven efficacy against influenza. ddhCTP is an endogenously produced nucleotide analog and chain terminator, and we compared it to its 3’-dNTP homologue, 3’-dCTP. Furthermore, each of these nucleotide analogs have modification at specific position, i.e. either at the ribose or at the base, which helps to understand how the polymerase responds to each modification. We have added this sentence in introduction in l.83-84 for clarity:

“We have therefore compared several analogs of the same natural nucleotide to determine how the nature of the modifications changes selection/mechanism of action.”

Similarly, I do not see the benefits of adding cell experiments with three compounds and experiments with the nsp14 mutant to address proofreading because they were inconclusive.

While we acknowledge Reviewer #2 opinion, Reviewer #3 has a different opinion and strongly appraises the importance of these results:

“Interestingly, the ddhCTP didn't actually work in infected cells. However, the authors presented a few theories on why it didn't work and said they plan to follow up to elucidate why it didn't work in cells. I think those results will be very interesting for the larger community working in this area.”

We share the opinion of Reviewer #3 and have therefore decided to keep these results in the revised manuscript.

1) Abstract. The term "deep backtrack" appearing in the abstract is unclear. Moreover, I would also avoid the use of word "unambiguously" in the abstract. I guess a claim on something cannot be ambiguous.

We agree with Reviewer #2, and we have modified the abstract that reads now:

“We show that RDV incorporation does not terminate viral RNA synthesis, but leads the polymerase into backtrack as far as 30 nt,…”

2) The force at which MT experiments were done is important. Although mentioned in the Methods, the force should be also indicated in Figure 1A, and in the main text. This will help understand the assay and movement of the bead upon nucleotide incorporation. The mechanical characterization of the initial (ssRNA) and final (dsRNA) substrates should be included as supplementary.

We have included the force in Figure 1A caption, and in the main text l.97:

“… and at constant force, i.e. 35 pN if not mentioned otherwise.”

We have included the dsRNA force extension in Figure 1- Supplement 1B.

3) Experiments were done at 35 and 25 pN. These are far from physiological forces, likely to be near zero. How would this might affect the reported conclusions? It would be good to complement with some (bulk?) experiments done at zero or near zero force, to backup for instance some of the claims on product length and termination. Additionally, I expect the equivalent MT experiment at low force to result in the increase of bead vertical position due to the conversion of ssRNA to dsRNA. Did the authors try that?

We have answered the point about physiological force above. In short, in the nucleotide addition cycle of the coronavirus polymerase, the force only affects a large conformational change occurring after chemistry, i.e. after nucleotide selection and incorporation. Concerning measurements at very low force, the regime mentioned by Reviewer #2 is rarely explored in such experiment because such experiments would have to be performed well below ~10 pN (at which ssRNA and dsRNA are of the same extension, preventing any measurable change in extension from polymerase elongation). However, at such force, the noise would be too large and prevent the high spatiotemporal resolution investigation we present here.

4) Related to previous point, the fact that results were similar at 25 and 35 pN is not sufficient to claim that "tension does not play a significant role in RDV-TP incorporation". This claim is also repeated for other nucleotides analogues. Authors tested very particular conditions and such general conclusion cannot be extracted from that data.

We have answered this point above.

5) Protein purification. Biochemical characterization of the purified proteins should be included, for instance a SDS-PAGE with purified proteins.

We have added a gel in Figure 1 —figure supplement 1D.

6) Figure 2 —figure supplement 1 is cited before than Figure 2. Not very logical. Suggest perhaps including the structure of each compounds in the main figure where used?

While Reviewer #2 is correct in the order of the figures, we think it is better to have all the compounds in the same figure, for the sake of comparison.

7) Authors claim that their assay works at saturating NTP concentration, an issue they believe is problematic with other published works (Refs 17 and 21). Could the authors determine the ratio of incorporation of RDV-TP versus ATP to contrast with the other published works (Refs 17, 21)?

Reviewer #2 is correct that our data shows the ability of our assay to observe nucleotide analogs incorporation at saturating concentration of competing natural NTPs. We think our assay is complementary to ensemble approaches, and we have amended the discussion in this sense (see our answer to point 4.c of Reviewer #1).

As RDV-TP is not a terminator, we cannot measure its incorporation vs ATP, and therefore we do not claim we have such a number. We can only assume its incorporation probability is high as we still see its effect on the kinetics at 20 µM vs 500 mM ATP.

8) Authors claim incorporation of RDV-TP leads to backtracking, but not to proofreading. This is a bit confusing. What would it be the function of backtracking here?

This question is in line with questions (1) and (2) by Reviewer #1, which have been answered there. Backtracking may have other function than proofreading, potentially in similarity-assisted copy-choice RNA recombination. However, to date, the true function of backtracking in viral RdRp’s has not yet been elucidated.

(9) The paper would benefit of including a table summarizing the effect of the different NA in the polymerase activity, and pointing out the main conclusions for each compound.

We have now included such a table.

(10) The description of the model and the analysis of dwell time distributions is too technical and difficult to follow in its current form. Equations are not numbered, terminology not defined, a figure is embedded within the text… I find all these developments more appropriate for a more specialized journal.

We thank Reviewer #2 for pointing out the missing equation numbers. We have now included them and moved the embedded figure into the Supplementary Figure.

(11) I guess the experiments using cells infected with SARS-CoV2 required a special biohazard security lab and specific measures. Please indicate all these details.

We have amended the Materials and methods l.615-618 that includes now:

“All experiments involving live SARS-CoV-2 were carried out under biosafety level 3 (BSL-3) containment by personnel wearing the appropriate PPE, including powered air purifying respirators with Tyvek suits, aprons, booties, and double gloves.”

(12) I do not see the benefit of including these cell experiments (Figure 6 – S3-S4), in their present form. Perhaps, a proper motivation is missing. The experiments with the nsp14 mutant are not conclusive and do not see the reason to include them.

We answered this point above.

Reviewer #3:This manuscript focuses on understanding the mechanism of action of remdesivir in the inhibition of SARS-Cov2 polymerase, using single molecule methods. The findings are highly original, significant and surprising. The approach is highly robust and supported by a range of orthogonal studies. Overall, these findings should help those engaged directly in drug discovery by providing a critical foundational understanding for the action of remdesivir.The research described in this manuscript has several findings that significantly impact the broader field polymerase inhibition. First, the authors were able to show using single molecule methods that remdesivir-TP incorporation leads to polymerase backtrack. This is important because the pause is long enough that an ensemble assay could mistake this backtrack for a termination event. Secondly, the researchers found the effective incorporation of remdesivir-TP was determined by its absolute concentration. This suggests remdesivir-TP and similar nucleotide analogs incorporate via the SNA or VSNA pathway and would be more likely to add to the RNA chain when substrate concentration is low (independent of stoichiometry with the competing native nucleotide). Thirdly, the researchers found the effective incorporation rate of obligatory terminators was affected by the stoichiometry of their competing native nucleotide rather than their absolute concentration. This suggests that obligatory terminators are incorporated via the NAB pathway. The pausing that the researchers observed in the polymerase elongation kinetics have recently been demonstrated by two other groups. However, this study improved upon the assay conditions used by other researchers to recapitulate in vivo conditions and remove bias from kinetics measurements.The authors highlighted the issues with remdesivir, tested other nucleotide analogs, and proposed a better alternative based on their assays (ddhCTP). Interestingly, the ddhCTP didn't actually work in infected cells. However, the authors presented a few theories on why it didn't work and said they plan to follow up to elucidate why it didn't work in cells. I think those results will be very interesting for the larger community working in this area. It's clear that the authors made a substantial enough contribution on the mechanism of inhibition of SARS Cov2 polymerase to merit publication in eLife, independent of the work on the "improved" antiviral candidate.It would have been useful to clarify for the reader the pharmaceutical import of the putative delayed chain termination (or pausing) relative to actual chemical chain termination. In other words, I'm assuming that in both cases the viral genome is considered to be non-transcribed (in that a chemical agent has been incorporated into the growing strand). This is true for most compounds in this broad class of anti-virals. The issues are usually surrounding the width of the therapeutic index and the degree to which resistant mutants arise.

Coronaviruses are unique among positive-strand RNA viruses in that they encode a proofreading exonuclease. Although it is unclear how the polymerase and exonuclease activities are coordinated, the current assumption is that errors are recognized when located at the terminus of nascent RNA. Therefore, nucleotide analogues which manifest their antiviral activity when embedded in nascent RNA should evade excision by the exonuclease.

We have added text conveying this sentiment here in l.70:

“The latter proofreads the terminus of the nascent RNA following synthesis by the polymerase and associated factors, a unique feature of coronaviruses relative to all other families of RNA viruses.”

And in lines 75-77:

“In other words, nsp14 adds another selection pressure on NAs: not only they must be efficiently incorporated by nsp12, they must also evade detection and excision by nsp14.”

Overall, this manuscript constitutes a major advance in our understanding of chain termination in polymerases, and provides deep insights into the mechanism of action of remdesivir, which may contribute to further drug discovery efforts targeting this polymerase. Additionally, the authors have highlighted and addressed issues in the methodologies of previous mechanistic studies that led others to erroneous conclusions.

We thank Reviewer #3 for her/his strong appraisal of our work.